# Stochastic Re-weighted Gradient Descent via Distributionally Robust Optimization

**Ramnath Kumar** *ramnathk@google.com*
*Google Inc.*

**Kushal Majmundar** *majak@google.com*
*Google Inc.*

**Dheeraj Nagaraj** *dheerajnagaraj@google.com*
*Google Inc.*

**Arun Sai Suggala** *arunss@google.com*
*Google Inc.*

**Reviewed on OpenReview:** *https://openreview.net/forum?id=KCf5CLAXZq*

## Abstract

We present Re-weighted Gradient Descent (RGD), a novel optimization technique that improves the performance of deep neural networks through dynamic sample re-weighting. Leveraging insights from distributionally robust optimization (DRO) with Kullback-Leibler divergence, our method dynamically assigns importance weights to training data during each optimization step. RGD is simple to implement, computationally efficient, and compatible with widely used optimizers such as SGD and Adam. We demonstrate the effectiveness of RGD on various learning tasks, including supervised learning, meta-learning, and out-of-domain generalization. Notably, RGD achieves state-of-the-art results on diverse benchmarks, with improvements of $+0.7\%$ on DomainBed, $+1.44\%$ on tabular classification, $+1.94\%$ on GLUE with BERT, and $+1.01\%$ on ImageNet-1K with ViT.

## 1 Introduction

Deep neural networks (DNNs) have become essential for solving a wide range of tasks, including image classification, object detection, machine translation, and speech recognition. The most commonly used paradigm for learning DNNs is empirical risk minimization (ERM Vapnik (1999)), which aims to identify a network that minimizes the average loss of training data points. Several algorithms, including SGD (Nemirovsky et al., 1983), Adam (Kingma & Ba, 2015), and Adagrad (Duchi et al., 2011), have been proposed for solving ERM. However, a drawback of ERM is that it weighs all the samples equally, often ignoring the rare and more difficult samples and focusing on the easier and abundant samples. This leads to suboptimal performance on unseen data, especially when the training data is scarce (Namkoong & Duchi, 2017). Consequently, recent works have developed data re-weighting techniques for improving the performance of ERM. One particularly fruitful approach in this line of work is the framework of Distributionally Robust Optimization (DRO) (Ben-Tal et al., 2013), which assigns higher weights to hard examples, often leading to models with better performance than ERM.

DRO selects the best model while also accounting for various uncertainties in the training data distribution (Ben-Tal et al., 2013). In particular, DRO treats the data distribution as uncertain and finds models which are robust to perturbations in the data distribution (e.g., removing a small fraction of points, adding random noise to each data point, etc.). This makes the model more robust to noise in the training dataset. For example, in the context of classification, this forces the model to place less emphasis on noisy features

and more emphasis on useful and predictive features. Consequently, models optimized using DRO have good generalization guarantees on unseen samples, and good performance on heterogeneous subpopulations in the data (Namkoong & Duchi, 2017; Duchi & Namkoong, 2018).

Concretely, let $\Theta$ be the model space, $P_{\text{data}}$ be the data distribution, and $\ell(\theta, z)$ be the loss of point $z$ w.r.t model $\theta$. Unlike ERM which minimizes the average loss $\mathbb{E}_{z \sim P_{\text{data}}}[\ell(\theta, z)]$, DRO minimizes the following objective

$$\inf_{\theta} \sup_{P':D(P'||P_{\text{data}}) \leq \rho} \mathbb{E}_{P'}[\ell(z, \theta)].$$

Here, $\rho$ is the perturbation radius, and $D$ is a divergence that measures the distance between two probability distributions. Popular choices for $D$ include $f$-divergences, which are defined as $D_f(Q||P) = \mathbb{E}_P[f(dQ/dP)]$ for some convex function $f : \mathbb{R}_+ \to \mathbb{R}$. In this case, Shapiro (2017) derived the following *equivalent* dual formulation of the above objective

$$\inf_{\lambda \geq 0} \inf_{\eta \in \mathbb{R}} \mathbb{E}_{z \sim P_{\text{data}}} \left[ \lambda f^* \left( \frac{\ell(\theta, z) - \eta}{\lambda} \right) \right] + \lambda \rho + \eta. \tag{1}$$

Here, $f^*(s) = \sup_t \{st - f(t)\}$ is the Fenchel conjugate of $f$. This alternative way of expressing DRO shows how it implicitly reweighs samples using the conjugate $f^*$. The seminal works of Duchi et al. (2021a); Namkoong & Duchi (2017) studied this objective for various $f$-divergences and showed that it has variance reduction properties, and leads to models with good generalization performance under *small* perturbations ($\rho = O(1/n)$, where $n$ is the data size). Furthermore, Duchi & Namkoong (2018) showed that DRO under *large* perturbations ($\rho = O(1)$) leads to models with good fairness, tail risk guarantees. In another line of work, Li et al. (2021) (and its extended version Li et al. (2023)) considered KL divergence DRO, which is obtained by choosing $f(x) = x \log x$, and showed that setting dual variable $\lambda$ to a negative value results in robust models that can withstand corruptions in the training data.

Inspired by these impressive properties, several recent studies have developed algorithms for optimizing DRO and designed data re-weighting techniques for various learning tasks. These algorithms fall into two broad categories: (a) *Primal-Dual techniques* which rely on alternating mirror ascent, descent to solve the min-max DRO objective (Namkoong & Duchi, 2016; Yan et al., 2020b; Fidon et al., 2020), and (b) *Compositional optimization techniques* which solve an equivalent compositional/dual form of DRO, which takes the form $g(\mathbb{E}_z[h(z, \theta)])$, for some functions $g, h$ (Qi et al., 2021; 2023b;a; Li et al., 2023). While these algorithms come with good convergence guarantees, they have certain drawbacks that limit their use in practice. (a) *Scalability:* primal-dual algorithms require updating and sampling from a probability distribution over the entire dataset at each iteration, making them computationally expensive. Although compositional optimization techniques alleviate this issue, gradient estimation within these algorithms is non-trivial as the objective is no longer an empirical mean of the losses evaluated at the training data points. Overcoming this often necessitates maintaining moving averages of sample weights, which introduces additional hyperparameters, complicating their application to large-scale scenarios (Qi et al., 2023b; Li et al., 2021). (b) *Robustness to Outliers:* many real-world datasets contain outliers[1], which pose challenges to algorithms optimizing DRO. In particular, these outliers often result in poor performance and instability during the DRO training process (Zhu et al., 2022a; Zhai et al., 2021). Existing works often fail to account for outliers in real datasets, leading to subpar performance (see Table 1 for a detailed comparison).

In this work, we address the aforementioned limitations of DRO optimization techniques. We focus on KL divergence-based DRO, and develop a lightweight algorithm for efficiently solving the resulting objective. Our algorithm simply optimizes the inner objective in Equation 1 using SGD. This gives rise to our stochastic Re-weighted Gradient Descent (RGD) algorithm, a variant of the classical SGD, that re-weights data points during each optimization step based on their difficulty. A key component of our algorithm is weight clipping that we introduce to protect against (benign) outliers and stabilize the algorithm. As demonstrated in our experiments (see Section 4, Appendix D), weight clipping significantly improves the performance of our algorithm on numerous learning tasks involving real-world datasets. Another noteworthy aspect of our

---

[1]An outlier is a data point that lies significantly outside the typical pattern of a dataset. These outliers could be because of noise in data collection process or could be introduced by a malicious adversary. In this work, we are primarily concerned about the former type of outliers.

| Paper | Algorithm | Per-step compute complexity (compared to SGD) | Hyper-parameters | Handles outliers in DRO? |
|---|---|---|---|---|
| Namkoong & Duchi (2016) | P-D | $O(\log n)\times$ more expensive | lr of dual variables | No |
| Qi et al. (2021) | C-M | $2\times$ more expensive | moving avg. parameter | No |
| Li et al. (2021; 2023) Qi et al. (2023b) | C-M | same as SGD | exponential scale, moving avg. parameter | No[2] |
| **This Work** | stochastic optimization of inner obj. | same as SGD | clipping level | Yes |

Table 1: Comparison with relevant prior works for optimizing KL-DRO. See Section 2 for a detailed discussion. P-D in the $2^{nd}$ column refers to primal-dual, and C-M refers to compositional minimization. $3^{rd}$ column corresponds to the cost of running each step of the algorithm. $4^{th}$ column corresponds to additional parameters introduced by the algorithm on top of learning rate (lr) of primal variables.

algorithm is that it has the same runtime as SGD, has only one hyper-parameter, and scales to models with billions of parameters.

## 1.1 Evaluation

In our experiments, we show that using our re-weighting scheme on top of existing learning algorithms improves their generalization performance in a variety of learning tasks including supervised learning, meta learning, out-of-domain generalization. While prior works focused on settings involving fairness, class imbalance to show improvements of DRO type methods, our work is the first to show significant improvements in generalization in large scale learning tasks across various domains.

**Supervised Learning:** We evaluate RGD on several supervised learning tasks in language and vision domains. In the language domain, we apply RGD for BERT fine-tuning on the General Language Understanding Evaluation (GLUE) benchmark and show that RGD outperforms the BERT baseline by +1.94%. In the vision domain, we apply RGD for ImageNet-1K classification using ViT-S model, and show that RGD outperforms the ViT-S baseline by +1.01%.

**Tabular Classification:** Recently, Majmundar et al. (2022) introduced a tabular representation learning method called MET. Deep learning methods trained with the learned representations from MET achieved SOTA performance on downstream classification tasks, significantly improving upon Gradient Boosting decision trees (GBDT; Friedman (2001)). Our experiments show that applying RGD to the MET framework improves its performance by 1.51% and 1.27% on binary and multi-class tabular classification, respectively.

**Domain Generalization:** In domain generalization, the distributions of train and test datasets could be different (for example, training on pictures of real dogs and evaluating cartoon dogs). This task requires robustness to distribution shifts and DRO is a natural framework in this context. Gulrajani & Lopez-Paz (2021) showed that the ERM framework applied over deep networks, is highly effective for this problem. Perhaps surprisingly, this remained the state-of-the-art (SOTA) algorithm for a long time. Only recently, ERM has been beaten on this challenging task (Cha et al., 2022; Addepalli et al., 2023). In this work, we show that using RGD on top of these recent techniques further boosts their performance by 0.7% and gives SOTA results on this task.

**Meta-Learning:** In meta-learning, we aim to learn models that generalize to new tasks with limited data. Predominant approaches in this domain use the classical ERM to optimize deep networks (Finn et al., 2017; Snell et al., 2017; Kumar et al., 2023). However, a common issue in this domain is that the learned models solve most tasks but fail catastrophically in some tasks. Consequently, this has promoted works that focus on worst-case performance (Collins et al., 2020). Recently, Li et al. (2023) used KL-DRO to tackle this problem. However, the authors applied their algorithm for solving meta-regression on a toy-dataset that is free of outliers, and haven't showcased their algorithm on practically relevant datasets such as Omniglot, *mini*ImageNet. In this work, we show that using RGD as an off-the-hat addition to Model-Agnostic Meta-

Learning (MAML) ([Finn et al., 2017](#)) can significantly improve the worst-case accuracy of these models on meta-classification benchmarks such as Omniglot, *mini*ImageNet by up to 3%.

## 1.2 Contributions

This work makes the following key contributions:

- **KL-DRO Inspired Re-weighting (RGD).** We introduce RGD, a novel, lightweight data re-weighting technique that improves the generalization of deep neural networks. Inspired by the principles of KL-DRO, RGD dynamically re-weights samples during optimization based on their difficulty. We further enhance robustness with weight clipping to mitigate the influence of outliers. RGD is versatile and easily integrated with widely used optimizers like Adam, SGD.

- **State-of-the-Art (SOTA) Performance Enhancement.** Extensive experiments demonstrate that RGD delivers significant performance gains across diverse learning tasks. In tabular classification, RGD boosts the accuracy of MET ([Majmundar et al., 2022](#)) by +1.44%. For out-of-domain generalization, RGD outperforms FRR ([Addepalli et al., 2023](#)) on DomainBed by +0.7%. Additionally, RGD improves the performance of BERT on GLUE benchmarks by +1.94% and ViT on ImageNet-1K by +1.01%. (see Section [4](#), Appendix [D](#))

## 2 Related Work

This section reviews relevant research on DRO. For a comprehensive overview of other popular data reweighting methods in machine learning, including AdaBoost, curriculum learning, please refer to Appendix [A](#).

DRO dates back to the early works of [Ben-Tal et al.](#) ([2009](#); [2013](#)). Since then several works have studied various statistical and optimization aspects of DRO. The seminal works of [Lam](#) ([2016](#)); [Namkoong & Duchi](#) ([2017](#)); [Duchi et al.](#) ([2021b](#)) formally showed that minimizing empirical DRO risk - under *small* perturbations ($\rho = O(1/n)$, where $n$ is the data size) - is equivalent to minimizing sum of empirical risk and its standard deviation. Consequently, optimizing DRO leads to a better bias-variance trade-offs and generalizing models. In this work, we rely on this property of DRO to develop our re-weighting scheme. In another seminal work, [Duchi & Namkoong](#) ([2018](#)) showed that DRO - under *large* perturbations ($\rho = O(1)$) - leads to models with good tail performance.

The aforementioned properties of DRO has led to numerous works applying it in various learning scenarios. For instance, [Duchi & Namkoong](#) ([2018](#)); [Sagawa* et al.](#) ([2020](#)); [Qi et al.](#) ([2021](#); [2023b](#)); [Li et al.](#) ([2021](#); [2023](#)) used DRO to tackle problems of class-imbalanaced classification and fairness. In another line of work, [Namkoong & Duchi](#) ([2017](#)); [Fidon et al.](#) ([2020](#)) studied DRO for designing models that generalize better than ERM. Our work falls in this second category of works that focus on generalization.

From an optimization perspective, several works have focused on designing efficient algorithms for optimizing the DRO objective. These algorithms can be classified into two broad categories: *primal-dual* ([Namkoong & Duchi](#), [2016](#); [Fidon et al.](#), [2020](#); [Yan et al.](#), [2020b](#)), *compositional optimization techniques* ([Qi et al.](#), [2021](#); [2023b](#);[a](#); [Li et al.](#), [2021](#)). One of the key drawbacks of primal-dual techniques is that they update and sample from a probability distribution over the entire training data at each step (*aka.* dual variables). A naive implementation of this step takes $O(n)$ time, which is prohibitive for large-scale tasks. [Namkoong & Duchi](#) ([2016](#)) reduced the complexity of this step to $O(\log n)$ using data structures such as balanced binary search trees. However, the resulting algorithms are hard to implement in practice. Another drawback of these algorithms is that they require storing a buffer of weights for the entire dataset, which is infeasible at the scale of LLMs. Even if storing the weights is feasible, the presence of data augmentations complicates the resulting algorithms. Compositional optimization algorithms overcome these drawbacks by working with an equivalent dual formulation of DRO that can be written as composition of two functions: $g(\mathbb{E}_z[h(z, \theta)])$. One major drawback of these techniques though is that estimating the gradient $\nabla_\theta g(\mathbb{E}_z[h(z, \theta)])$ from a mini-batch is

---

[2][Li et al.](#) ([2021](#)) developed hierarchical TERM framework for handling outliers in DRO. But their framework is only applicable to settings such as group fairness where the learner has a priori knowledge of group memberships.

non-trivial. This requires certain additional steps in the algorithm which add to its complexity. For instance, the algorithm of Qi et al. (2021) requires making two backward passes at each step of SGD. The convergence analysis of Li et al. (2021) required two independent mini-batches at each iteration. The algorithms of Qi et al. (2023b); Li et al. (2021) both require maintaining a moving average of the weights of the mini-batches, thus adding an additional hyper-parameter to the algorithm.

**Outlier robust DRO.** Recent works have shown that DRO is highly sensitive to outliers (Zhai et al., 2021; Zhu et al., 2022a). This is because DRO tends to magnify the influence of outliers by upweighting them further. This is one of the reasons for the suboptimal performance of existing DRO algorithms on real world datasets. To address this, Zhai et al. (2021) consider an adversarial model for outliers (where $\epsilon$-fraction of training data could be arbitrarily corrupted by a malicious adversary). They develop a heuristic to remove the outliers during each descent step for $\chi^2$-DRO and CVaR. While interesting, this is too strong of an adversary model which leads to data wastage in practice.

**Min-min DRO.** Min-min DRO minimizes the following objective: $\inf_{P':D(P'||P_{\text{data}})\leq\rho} \mathbb{E}_{P'}[\ell(z,\theta)]$. Contrast this with DRO which minimizes: $\sup_{P':D(P'||P_{\text{data}})\leq\rho} \mathbb{E}_{P'}[\ell(z,\theta)]$. Unlike DRO which is primarily studied for generalization and fairness properties, min-min DRO is studied for training in the presence of outliers (Li et al., 2021; Kumar & Amid, 2021; Majidi et al., 2021). Instead of upweighting high loss points, min-min DRO downweights them.

**Other applications of DRO.** Sagawa* et al. (2020) optimized Group DRO for fair models when the group information is known. Sinha et al. (2018) studied DRO with Wasserstein divergence for learning models that are robust to adversarial perturbations. DRO also appears in many classical statistical problems. For example, many boosting algorithms (including AdaBoost) can be viewed as performing DRO with KL-divergence-based uncertainty sets (Arora et al., 2012; Friedman, 2001). Faury et al. (2020) relied on KL-DRO for counterfactual risk minimization. Sakhi et al. (2020) use DRO for improving offline contextual bandits algorithms.

**Optimization Techniques for Improved Generalization.** Several optimization techniques, that fall outside the umbrella of DRO, have been proposed for improving the generalization of ML models. Of these, Sharpness-Aware Minimization (SAM) (Foret et al., 2021) is perhaps the most popular technique. From a theoretical perspective, SAM performs robust optimization in the weight space (that is SAM tries to learn a model that is robust to perturbations of weights). In contrast, RGD performs robust optimization in the distribution space. So, RGD and SAM are orthogonal to each other and can potentially be merged together to boost the performance. See Appendix D.3 for empirical comparison between RGD and SAM.

## 3 Algorithm and Derivation

In this section, we first formally introduce DRO and describe its generalization properties. Next, we derive RGD as a technique for solving DRO.

### 3.1 Distributionally Robust Optimization

Consider a general learning problem where we are given $n$ i.i.d samples $\{z_i\}_{i=1}^n$ drawn from some unknown distribution $P_{\text{data}}$. Let $\widehat{P}_{\text{data}}$ be the empirical distribution over these samples. Our ideal goal is to find a model $\theta \in \Theta$ that minimizes the population risk: $R(\theta) \coloneqq \mathbb{E}_{P_{\text{data}}}[\ell(z;\theta)]$. Here $\ell(z;\theta)$ is the loss of $z$ under model $\theta$. Since $P_{\text{data}}$ is typically unknown, a standard practice in ML/AI is to minimize the empirical risk, which is defined as

$$\widehat{R}_n(\theta) \coloneqq \mathbb{E}_{\widehat{P}_{\text{data}}}[\ell(z;\theta)] = \frac{1}{n}\sum_{i=1}^n \ell(z_i;\theta).$$

In DRO, we assume that a "worst-case" data distribution shift may occur, which can harm a model's performance. So, DRO optimizes the loss for samples in that "worst-case" distribution, making the model robust to perturbations (see Figure 2 for illustration). Letting $D$ be a divergence that measures the distance

between two probability distributions, the population and empirical DRO risks w.r.t $D$ are defined as

$$R_D(\theta) \coloneqq \sup_{P':D(P'||P_{\text{data}})\leq\rho} \mathbb{E}_{P'}[\ell(z,\theta)], \quad \widehat{R}_{D,n}(\theta) \coloneqq \sup_{P':D(P'||\widehat{P}_{\text{data}})\leq\rho} \mathbb{E}_{P'}[\ell(z,\theta)].$$

Here, $\rho$ is the perturbation radius. Popular choices for $D$ include $f$-divergences, which are defined as $D_f(Q||P) = \mathbb{E}_P[f(dQ/dP)]$ for some convex function $f : \mathbb{R}_+ \to \mathbb{R}$. We note that many popular divergences, such as Kullback–Leibler (KL) divergence ($f(x) = x\log x$), Total Variation distance ($f(x) = \frac{1}{2}|x-1|$), and $\chi^2$-divergence ($f(x) = (x-1)^2$), fall into this category.

**Generalization.** Models learned using ERM can suffer from poor generalization (*i.e.,* performance on unseen data) in high-variance settings. For instance, consider the following well-known generalization guarantee that holds with high probability for any $\theta \in \Theta$ (Wainwright, 2019)

$$R(\theta) \leq \widehat{R}_n(\theta) + c_1\sqrt{\frac{\text{Var}_{\widehat{P}_{\text{data}}}(\ell(z;\theta))}{n}} + \frac{c_2}{n}. \tag{2}$$

Here, $c_1, c_2 > 0$ are constants, and $\text{Var}_P(\ell(z;\theta))$ is the variance of $\ell(z;\theta)$ w.r.t distribution $P$. Such bounds hold under certain regularity conditions on $\ell$ and $\Theta$. While ERM minimizes the first term in the RHS above, it totally ignores the second term involving the variance. Consequently, in high-variance and/or small $n$ settings where $R(\theta)$ and $\widehat{R}_n(\theta)$ are far away from each other, ERM tends to have poor generalization guarantees. A natural technique to address this issue is to learn models that consider the bias-variance trade-off and minimize the following objective.

$$\min_{\theta\in\Theta} \widehat{R}_n(\theta) + c_1\sqrt{\frac{\text{Var}_{\widehat{P}_{\text{data}}}(\ell(z;\theta))}{n}}.$$

However, minimizing this objective is computationally intractable even when the loss $\ell$ is convex in $\theta$, as the overall objective is non-convex. Recent works have made an interesting connection between the above objective and DRO to address this issue. Specifically, when $D$ is an $f$-divergence, the following result holds with high probability, whenever the perturbation radius $\rho = \frac{c}{n}$, for some appropriately chosen constant $c > 0$ (Lam, 2019; Namkoong & Duchi, 2017)

$$\widehat{R}_{D,n}(\theta) = \widehat{R}_n(\theta) + c_1\sqrt{\frac{\text{Var}_{\widehat{P}_{\text{data}}}(\ell(z;\theta))}{n}} + \frac{c_3}{n} \quad \forall\theta\in\Theta.$$

Ignoring the lower order terms (i.e., $1/n$ terms), the above equation, together with Equation 2, shows that the empirical DRO risk $\widehat{R}_{D,n}(\theta)$ is an upper bound of the population risk $R(\theta)$ at any $\theta \in \Theta$. Furthermore, it can be seen that empirical DRO is equal to the empirical risk plus a variance term (modulo the lower order term). This variance term acts as a regularizer during the optimization of empirical DRO and leads to models with smaller variance, and with good generalization guarantees.

### 3.2 Stochastic Re-weighted Gradient Descent (RGD)

The above discussion motivates the use of DRO risk for learning models, especially in high variance and/or low sample regime. We now derive our RGD algorithm as a technique to minimize the empirical DRO risk $\widehat{R}_{D,n}$, and to learn models with better generalization guarantees than ERM. Specifically, we consider KL divergence-based DRO, where one adds perturbations to create data distributions that are close to the original data distribution in the KL divergence metric, and learn a model with best performance over all possible perturbations. The following proposition derives the *equivalent* dual representation of the KL divergence-DRO objective, the proof of which can be found in Appendix B.1.

**Proposition 3.1.** *(Shapiro, 2017) Consider DRO with KL-divergence-based uncertainty set. Then* $\min_{\theta\in\Theta}\widehat{R}_{D,n}$ *can be rewritten as:*

$$\min_{\theta\in\Theta} \frac{1}{\gamma}\log\mathbb{E}_{\widehat{P}_{data}}[e^{\gamma\ell(z;\theta)}],$$

*for some constant $\gamma > 0$ that is independent of $\theta$.*

---

**Algorithm 1** Re-weighted Gradient Descent (RGD)

---

1: **Input:** Data $\{z_i\}_{i=1}^n$, learning rate sequence $\{\eta_t\}_{t=1}^T$, number of iterations $T$, loss function $\ell$, re-weighting function $g$, mini-batch size $B$
2: **for** $t = 0 \ldots T - 1$ **do**
3:     Sample minibatch $\{z_i\}_{i=1}^B$
4:     Compute losses for points in the minibatch:

$$\ell_i \leftarrow \ell(z_i; \theta_t), \ \forall i \in 1 \ldots B$$

5:     Compute per-sample weights:

$$w_i \leftarrow g(\ell_i) \ \forall i \in 1 \ldots B$$

6:     Compute the weighted pseudo-gradient:

$$v_t \leftarrow \frac{1}{B} \sum_{i=1}^B w_i \nabla_\theta \ell(z_i; \theta_t)$$

7:     Update model parameters:

$$\theta_{t+1} \leftarrow \Pi_\Theta(\theta_t - \eta_t v_t)$$

8: **end for**

---

Equipped with the dual representation, we now derive our RGD algorithm. Observe that minimizing the compositional objective $\log \mathbb{E}[\exp(\gamma \ell(z; \theta))]$ is equivalent to minimizing the inner objective $\mathbb{E}[\exp(\gamma \ell(z; \theta))]$. In this work, we perform SGD on this inner objective, which leads to the following update:

$$\theta_{t+1} \leftarrow \Pi_\Theta \left( \theta_t - \gamma \eta_t \frac{1}{B} \sum_{i-1}^B e^{\gamma \ell(z_i; \theta_t)} \nabla_{\theta_t} \ell(z_i; \theta_t) \right).$$

Here $\Pi_\Theta$ is the projection onto the feasible set $\Theta$. The following proposition shows that this update rule converges to the minimum of $\widehat{R}_{D,n}$ under certain conditions on $\ell(z;)$. The proof, given in Appendix B.2 follows from an application of Shamir & Zhang (2013, Theorem 2).

**Proposition 3.2.** *Assume that $\Theta$ is a convex and compact set. For all data points $z$ suppose that $\ell(z; \cdot)$ is convex, continuously differentiable and bounded in the range $[-M, M]$, and $\nabla \ell(z; \cdot)$ is uniformly bounded over the set $\Theta$. Let the step-size sequence $(\eta_t)_t$ be such that $\eta_t = \frac{C}{\sqrt{t}} \ \forall \ 1 \le t \le T$ or $\eta_t = \frac{C}{\sqrt{T}} \ \forall \ 1 \le t \le T$. Then the sub-optimality gap satisfies:*

$$\mathbb{E}_{\theta_T} \frac{1}{\gamma} \log \mathbb{E}_{\widehat{P}_{data}}[e^{\gamma \ell(z; \theta_T)}] - \min_{\theta \in \Theta} \frac{1}{\gamma} \log \mathbb{E}_{\widehat{P}_{data}}[e^{\gamma \ell(z; \theta)}] = O\left(\frac{\log T}{\sqrt{T}}\right).$$

This shows that choosing the re-weighting function $g(u)$ in Algorithm 1 as $e^{\gamma u}$ (for some appropriate choice of $\gamma$) leads to robust models with better generalization guarantees. This is the choice of $g$ we use in our paper. One thing to note here is that, even though the proposition is specific to SGD, it can be easily extended to other optimization techniques such as Adam.

**Weight Clipping.** In our experiments, when computing per-sample weights, we clip the loss $\ell$ at some constant $\tau > 0$; that is, we use $g(u) = e^{\gamma \min(u, \tau)}$. We observed this clipping to help stabilize the training in the presence of outliers (see Section 4.1 for empirical evidence). We note that this is different from other common techniques, such as loss, gradient clipping, which are used to make the learning process robust to outliers in empirical risk minimization (Yang et al., 2010; Catoni & Giulini, 2018; Menon et al., 2019; Koloskova et al., 2023). We plan to investigate the robustness properties of weight clipping in more detail in future work. In our experiments, we choose the scale parameter $\gamma = 1/(\tau + 1)$. Even with this fixed choice of $\gamma$, our algorithm provides a significant boost in performance over vanilla optimization techniques (see Section 4.1 for ablations on our design choices).

**Choice of divergences.** One could rely on other divergences instead of the KL-divergence we used in the above derivation. From a theoretical perspective, DRO with many $f$-divergences (KL, reverse KL, chi-squared etc.) provides an upper bound on the true population risk (Lam, 2016; Duchi et al., 2021b). For example, using $\chi^2$-divergence ($f(x) = (x-1)^2$) gives us the following re-weighting function for positive loss functions: $g(u) = u + \tau$. Using reverse KL-divergence ($f(x) = -\log x$) gives us the following re-weighting function $g(u) = \frac{1}{\tau - u}$ for some appropriate choice of $\tau$ (see Appendix B.3 for more details). Observe that the reverse KL-divergence based weighting function is much more aggressive in up-weighting high loss points than KL-divergence based weighting function. In the sequel, we denote the approach with $g(u) = u + \tau$ as RGD-$\chi^2$, $g(u) = \frac{1}{\tau - u}$ as RGD-REVKL, and $g(u) = e^{\min(u,\tau)/(\tau+1)}$ as RGD. While all the three techniques provided a performance boost over ERM (see Appendix C), KL-divergence based reweighting has better performance than reverse KL, chi-squared divergences. However, the current theoretical understanding of DRO doesn't explain this nuanced behavior and we believe that this is an interesting direction for future research. From a practitioner perspective, the choice of $f$-divergence could be treated as a hyper-parameter which could be tuned using cross-validation.

## 4 Experiments

In this section, we first present ablations on various design choices in our algorithm. Next, we present empirical evidence showing that RGD outperforms ABSGD (Qi et al., 2023b), TERM (Li et al., 2021), two state-of-the-art algorithms for optimizing KL-DRO. Finally, we present large scale experiments showing that RGD can be widely applied across tasks such as supervised learning, meta learning to boost the generalization performance of existing learning algorithms. Details regarding hyperparameter tuning are relegated to Appendix 5. Additional experiments on class imbalanced classification, and large-scale tasks such as miniGPT pre-training, EfficientNet finetuning are presented in Appendix D.

To ensure fair comparisons, we integrated RGD into existing baseline codebase for each of the experiments, maintaining the same optimizer (primarily Adam for most experiments, SGD for CIFAR experiments) for both baseline and RGD versions. RGD introduces only one additional hyperparameter, the clipping factor ($\tau$). The optimizer, weight decay, batch sizes, etc. were kept constant with the baseline across all our experiments.

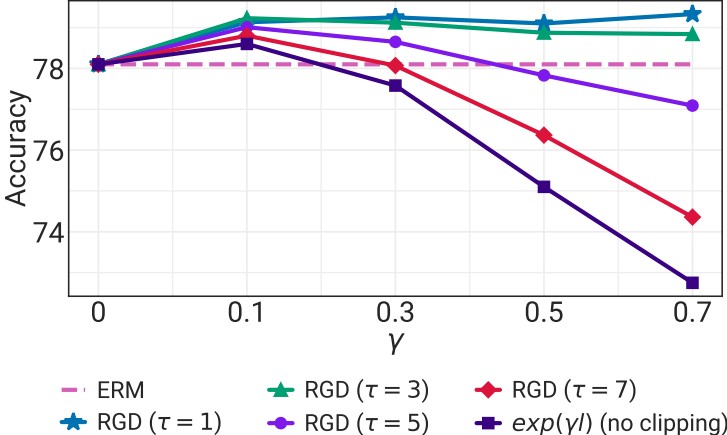

Figure 1: Ablation of scaling and clipping factor of RGD training regime on the Imagenet dataset with a ViT-S backbone.

### 4.1 Ablation Studies

In this section, we present ablations justifying the key design choices in our algorithm. All the results presented here are for ImageNet-1K classification using ViT-S model.

**Choice of scale parameter ($\gamma$).** Recall, RGD uses $\gamma = \frac{1}{\tau+1}$, where $\tau$ is the clipping level. To understand the utility of this choice, we vary $\gamma$. To be precise, we set $\tau = 1$ and get accuracy numbers of RGD with

various values of $\gamma$. Figure 1 (light blue line with *stars*) presents the results from this experiment. For each value of $\gamma$, we report the accuracy obtained using the best learning rate identified using hold-out set validation. It can be seen that RGD is fairly robust to the choice of $\gamma$.

**Importance of clipping.** We now study the importance of clipping. To this end, we replace the proposed reweighting function in RGD with $g(u) = e^{\gamma u}$. Figure 1 (dark blue line with *squares*) presents the results from this experiment, for various values of $\gamma$. It can be seen that the best accuracy without clipping is 1% off compared to RGD with clipping (with $\tau = 1$). We believe this performance drop primarily happens because of the high weights given to the outliers. This demonstrates the importance of clipping. Next, we vary the clipping factor $\tau$. It is evident that as $\tau$ increases, the performance of RGD drops and approaches the performance of no clipping. This shows the importance of setting an appropriate $\tau$.

To further demonstrate that our weight clipping can effectively handle benign outliers, we perform the following experiment. We randomly flip the labels in CIFAR-10, CIFAR-100 datasets (we vary the proportion of flips from 0% to 40%) and compare the performance of RGD with state-of-the-art KL-DRO optimization technique TERM (Li et al., 2021). The results from this experiment are reported in Table 2. Note that TERM, which does not account for outliers, performs poorly in this experiment, as it upweights the corrupted/mislabeled points. Whereas, RGD which clips the weights of outliers, achieves the best performance.

Table 2: Results on CIFAR-10 and CIFAR-100 dataset with corrupted labels.

| Dataset | CIFAR-10 | | | | CIFAR-100 | | | |
|---|---|---|---|---|---|---|---|---|
| Loss | 0% | 20% | 40% | **Avg.** | 0% | 20% | 40% | **Avg.** |
| Default (Cross Entropy) | $92.89_{\pm 0.32}$ | $76.83_{\pm 2.30}$ | $70.77_{\pm 2.31}$ | 80.16 | $70.50_{\pm 0.12}$ | $50.86_{\pm 0.27}$ | $43.01_{\pm 1.16}$ | 54.79 |
| TERM (Li et al., 2021) | $92.90_{\pm 0.09}$ | $58.7_{\pm 39.76}$ | $73.17_{\pm 31.59}$ | 74.92 | $70.42_{\pm 0.41}$ | $63.85_{\pm 1.40}$ | $46.59_{\pm 22.80}$ | 60.29 |
| RGD **(Ours)** | $\mathbf{93.04}_{\pm 0.24}$ | $\mathbf{90.69}_{\pm 0.24}$ | $\mathbf{88.90}_{\pm 0.15}$ | **90.88** | $\mathbf{71.06}_{\pm 0.22}$ | $\mathbf{64.61}_{\pm 0.39}$ | $\mathbf{57.17}_{\pm 0.80}$ | **64.28** |

## 4.2 Comparison with existing KL-DRO optimization techniques

In this section, we present experimental results on ImageNet classification with ViT-S model to showcase the efficacy of RGD over state-of-the-art KL-DRO optimization techniques ABSGD, and TERM (see Table 3). We were unable to compare with the recent SCDRO algorithm of Qi et al. (2023a) due to technical difficulties with implementing it in JAX (Bradbury et al., 2018) (the main difficulty arises from syncing values of parameters across various devices). Furthermore, the algorithm is significantly more complex with many hyperparameters. Table 3 shows that ABSGD, TERM with extensive tuning of three hyperparameters (exponential scale, learning rate , and moving average parameter) achieve similar performance as baseline ERM. In contrast, RGD outperforms the baseline ERM by 1.1% with minimal tuning of hyperparameters, highlighting its efficacy. One of the reasons for this performance difference between RGD, and ABSGD, TERM is the weight clipping we perform in our algorithm, which guards it from outliers. We note that SCDRO doesn't perform weight clipping and could potentially suffer from a similar performance drop as ABSGD, TERM. Additional details about the experiment can be found in Appendix D.1.1.

Next, we compare RGD with ABSGD, TERM for the problem of class imbalanced classification. For this experiment, we consider the long-tailed CIFAR-10, CIFAR-100 datasets (Cui et al., 2019). Table 4 presents the results from this experiment. It can be seen that RGD outperforms both ABSGD and TERM by $> 1\%$ on average. Additional details about this experiment, including comparison with specialized techniques for class imbalanced classification such as *class-balanced* loss (Cui et al., 2019) and *focal* loss (Lin et al., 2017), can be found in Appendix D.2.

Table 3: Comparison of RGD with existing algorithms for solving KL-DRO on ImageNet-1K classification with ViT-S model.

| **Algorithm** | ERM | TERM | ABSGD | RGD |
|---|---|---|---|---|
| Accuracy | $78.44_{\pm 0.35}$ | $78.03_{\pm 0.27}$ | $77.12_{\pm 0.82}$ | $\mathbf{79.11}_{\pm 0.12}$ |

Table 4: Comparison of RGD with existing KL-DRO optimization techniques for class imbalanced classification. The numbers represent test accuracy on Long-Tailed CIFAR-10, and CIFAR-100 datasets using ResNet-32. Moreover, we perform hypothesis tests to confirm that these results are statistically significant with a p-value that is less than 0.05.

| Dataset | CIFAR-10 | | | CIFAR-100 | | |
|---|---|---|---|---|---|---|
| Loss / Imbalance Factor | 100 | 10 | **Avg.** | 100 | 10 | **Avg.** |
| Cross Entropy (CE) | | | | | | |
| Default | $71.75 _{\pm 0.75}$ | $87.64 _{\pm 0.45}$ | 79.7 | $38.35 _{\pm 38.35}$ | $56.91 _{\pm 0.41}$ | 47.63 |
| TERM Li et al. (2021) | $72.20 _{\pm 0.18}$ | $87.52 _{\pm 0.07}$ | 79.86 | $39.75 _{\pm 0.04}$ | $57.91 _{\pm 0.39}$ | 48.83 |
| ABSGD Qi et al. (2023b) | $72.43 _{\pm 0.31}$ | $\mathbf{87.93} _{\pm 0.25}$ | 80.18 | $39.77 _{\pm 0.34}$ | $57.44_{\pm 0.25}$ | 48.61 |
| RGD **(Ours)** | $\mathbf{73.71} _{\pm 0.04}$ | $87.78 _{\pm 0.15}$ | **80.75** | $\mathbf{41.47} _{\pm 0.30}$ | $\mathbf{58.49} _{\pm 0.04}$ | **49.98** |

## 4.3 Supervised Learning

This section studies our approach when applied on standard supervised learning tasks such as BERT finetuning on GLUE benchmark, and Imagenet-1K classification. We use a base model of ViT-S for the latter task. Table 5 depicts our results from this experiment. On GLUE tasks, our RGD algorithm outperforms the baseline by **+1.94%** with a standard deviation of 0.42%. Furthermore, we perform hypothesis testing to condit that these results are statistically significant with a p-value that is less than 0.05. On Imagenet-1K, we show a **+1.01%** improvement over baseline with the off-the-hat addition of the RGD reweighing and no additional complexity in terms of compute: memory and time.

Furthermore, we also experiment with pre-training of the BERT-base model. We use the BooksCorpus (800M words) (Zhu et al., 2015) and English Wikipedia (2,500M words) as our pre-training corpus. We trained the Bert-base model for 450K steps, and tuned the learning rate (lr) for baseline, and lr, clipping factor for RGD. We report both the MLM (Masked Language Model) accuracy and NSP (Next Sequence Prediction) accuracy comparisons of RGD vs Default (ERM). It can be seen that our approach boosts the MLM accuracy and NSP accuracy by **+0.2%** and **+0.9%** respectively (see Table 5). Furthermore, through hypothesis testing, we show that the results are statistically significant with a p-value of 0.05. Additional experiments on EfficientNet fine-tuning, DeiT model (Touvron et al., 2021) for ImageNet-1K classification, and miniGPT (Zhu et al., 2024) pre-training are discussed in Appendix D.5.

Table 5: Performance of RGD for various Supervised Learning tasks.

| | MNLI | QQP | QNLI | SST-2 | MRPC | RTE | COLA | Avg on GLUE | ImageNet-1K | BERT Pretraining (MLM) | BERT Pretraining (NSP) |
|---|---|---|---|---|---|---|---|---|---|---|---|
| Default | 81.33 | 89.62 | 87.93 | 90.63 | **89.55** | 67.19 | 54.53 | 80.11 | $78.44 _{\pm 0.35}$ | 71.31 | 98.01 |
| RGD **(Ours)** | **83.06** | **91.06** | **90.35** | **91.78** | 88.28 | **71.48** | **58.56** | **82.05** | $\mathbf{79.11} _{\pm 0.12}$ | $\mathbf{71.35} _{\pm 0.11}$ | $\mathbf{98.92} _{\pm 0.09}$ |

## 4.4 Tabular Classification

Table 6: Results on standard binary-class tabular datasets (AUROC): The bottom partition shows results of our method with RGD loss. We show that the addition of our proposed approach significantly outperforms existing methods, as well as SOTA.

| Algorithm | Obesity | Income | Criteo | Thyroid | Avg. |
|---|---|---|---|---|---|
| MLP | 52.3 | 89.39 | 79.82 | 62.3 | 70.95 |
| RF Breiman (2001) | 64.36 | 91.53 | 77.57 | 99.62 | 83.27 |
| MET-S | | | | | |
| Default Majmundar et al. (2022) | 71.84 | 93.85 | 86.17 | 99.81 | 87.92 |
| RGD **(Ours)** | **76.87** | **93.96** | **86.98** | **99.92** | **89.43** |

Table 7: Results on standard multi-class tabular datasets (Accuracy): The bottom partition shows results of our method with RGD loss. We show that the addition of our proposed approach significantly outperforms existing methods, as well as SOTA.

| Algorithm | FMNIST | CIFAR10 | MNIST | CovType | Avg. |
|---|---|---|---|---|---|
| MLP | 87.62 | 16.50 | 96.95 | 65.47 | 66.64 |
| RF Breiman (2001) | 88.43 | 42.73 | 97.62 | 71.37 | 75.04 |
| MET Majmundar et al. (2022) | **91.68** | 47.82 | 99.19 | 76.71 | 78.85 |
| MET-S | | | | | |
| Default Majmundar et al. (2022) | 90.94 | 48.00 | 99.01 | 74.11 | 78.02 |
| RGD (**Ours**) | 91.54 | **49.54** | **99.69** | **79.72** | **80.12** |

Learning with tabular data is a task where traditional machine learning methods, like random forest Breiman (2001); Friedman (2001) are incredibly competitive against deep learning-based methods (Yoon et al., 2020). Recently, Majmundar et al. (2022) obtained SOTA results for tabular classification using self-supervised representation learning and relying on the learned representations in the downstream classification tasks (see Section 1). Their work developed two algorithms namely, MET (representation learning with adversarial training) and MET-S (representation learning without adversarial training). The adversarial training adds robustness to the learned representations, thus improving performance. In this experiment, we integrate RGD with MET-S instead of doing adversarial training. This allows us to test the robustness properties of the models trained with RGD. Table 7 and Table 6 shows gains on multiple tabular datasets for the multi-class classification and binary classification tasks. Notably, our approach outperforms previous SOTA in this problem by **+1.27%**, and **+1.5%** on the multi-class and binary classification tasks respectively. We refer to Appendix D.6 for a comprehensive comparison with baselines such as Gradient Boosting Decision Trees (Friedman, 2001), VIME (Yoon et al., 2020), SubTab (Ucar et al., 2021), TabNet (Arik & Pfister, 2021), DACL+ (Verma et al., 2021) and many more. Our motivation to experiment on these "permuted" MNIST, "permuted" CIFAR, and "permuted" FMNIST can be traced back to the introduction of these datasets in the works of Yoon et al. (2020); Ucar et al. (2021). Subsequently, other recent works such as Majmundar et al. (2022) also experimented on these datasets and have become a standard benchmark for tabular classification.

## 4.5 Out Of Domain Generalization

Table 8: Results on DomainBed (Model selection: training-domain validation set): The bottom partition shows results of our method with RGD loss. In both cases, with (top) and without (bottom) fixed linear layer, the proposed approach outperforms existing methods, as well as SOTA.

| Algorithm | PACS | VLCS | OfficeHome | DomainNet | Avg. |
|---|---|---|---|---|---|
| ERM Gulrajani & Lopez-Paz (2021) | $85.5 \pm 0.1$ | $77.5 \pm 0.4$ | $66.5 \pm 0.2$ | $40.9 \pm 0.1$ | 67.6 |
| MIRO Cha et al. (2022) | $85.4 \pm 0.4$ | **$79.0 \pm 0.0$** | $70.5 \pm 0.4$ | $44.3 \pm 0.2$ | 69.8 |
| ERM + FRR-L | | | | | |
| Default Addepalli et al. (2023) | $85.7 \pm 0.1$ | $76.6 \pm 0.2$ | $68.4 \pm 0.2$ | $44.2 \pm 0.1$ | 68.73 |
| RGD (**Ours**) | $87.6 \pm 0.3$ | $78.6 \pm 0.3$ | $69.8 \pm 0.2$ | **$46.00 \pm 0.0$** | 70.48 |
| ERM + FRR | | | | | |
| Default Addepalli et al. (2023) | $87.5 \pm 0.1$ | $77.6 \pm 0.3$ | $69.4 \pm 0.1$ | $45.1 \pm 0.1$ | 69.90 |
| RGD (**Ours**) | **$88.2 \pm 0.2$** | $78.6 \pm 0.3$ | $69.8 \pm 0.2$ | $45.8 \pm 0.0$ | **70.60** |

In this section, we show that our technique can be used to boost the performance of OOD generalization techniques. We experiment on DomainBed, a standard benchmark used to study the out-of-domain performance of models. More information about the benchmark, the task to solve, and the metric is discussed in Appendix D.8.1. The benchmark is notorious since the most basic approach, such as straightforward Empirical Risk Minimization (ERM) as evaluated by Gulrajani & Lopez-Paz (2021), was the SOTA method for a long time. Most new approaches either performed worse than ERM or marginally better. In recent

years, breakthroughs such as MIRO (Cha et al., 2022) and FRR (Addepalli et al., 2023) have pushed the problem further by significantly improving the benchmarks. We show that integrating our proposed approach RGD with these approaches (specifically FRR) significantly improves performance (an average of **+0.7%**). Table 8 illustrates the accuracy performance numbers of a few baseline methods and our proposed approach. A more comprehensive comparison with additional baselines such as IRM (Arjovsky et al., 2019), CORAL (Sun & Saenko, 2016), MTL (Blanchard et al., 2021), SagNet (Nam et al., 2021), and many more in the Table 24. We further depict the environment-wise breakdown of the accuracy of each of the baseline algorithms in Appendix D.8.

### 4.6 Meta-Learning

Table 9: Results on meta-learning datasets. We report the Worst-K% performance as well to help study the performance distribution over all tasks.

| Algorithm | Worst 10% | Worst 20% | Worst 30% | Worst 40% | Worst 50% | Overall |
|---|---|---|---|---|---|---|
| Omniglot 5-way 1-shot | | | | | | |
| MAML | $91.71 \pm 0.73$ | $94.16 \pm 0.50$ | $95.41 \pm 0.39$ | $96.22 \pm 0.32$ | $96.76 \pm 0.27$ | $98.38 \pm 0.17$ |
| MAML + RGD | $\mathbf{92.14} \pm 0.84$ | $\mathbf{94.54} \pm 0.53$ | $\mathbf{95.72} \pm 0.40$ | $\mathbf{96.46} \pm 0.33$ | $\mathbf{96.90} \pm 0.27$ | $\mathbf{98.45} \pm 0.17$ |
| Omniglot 20-way 1-shot | | | | | | |
| MAML | $84.33 \pm 0.40$ | $85.86 \pm 0.29$ | $86.92 \pm 0.26$ | $87.73 \pm 0.24$ | $88.42 \pm 0.22$ | $91.28 \pm 0.22$ |
| MAML + RGD | $\mathbf{86.61} \pm 0.36$ | $\mathbf{88.09} \pm 0.28$ | $\mathbf{89.09} \pm 0.24$ | $\mathbf{89.87} \pm 0.23$ | $\mathbf{90.50} \pm 0.21$ | $\mathbf{93.01} \pm 0.20$ |
| *mini*ImageNet 5-way 1-shot | | | | | | |
| MAML | $30.94 \pm 0.70$ | $34.52 \pm 0.62$ | $36.93 \pm 0.57$ | $38.94 \pm 0.55$ | $40.68 \pm 0.53$ | $48.86 \pm 0.62$ |
| MAML + RGD | $\mathbf{33.33} \pm 0.90$ | $\mathbf{36.67} \pm 0.65$ | $\mathbf{39.12} \pm 0.59$ | $\mathbf{41.20} \pm 0.56$ | $\mathbf{42.96} \pm 0.55$ | $\mathbf{51.21} \pm 0.63$ |

In meta-learning, the goal is to learn representations that generalize effectively to new tasks, even when provided with limited examples. However, task heterogeneity poses a significant challenge. Some tasks may be inherently simpler to learn, leading models to prioritize these and neglect the more difficult, less frequent tasks. While Empirical Risk Minimization (ERM) may perform well on common tasks, its performance can deteriorate drastically on rare and challenging ones. This necessitates a mechanism for re-weighting tasks to ensure balanced learning. Building upon the experimental results of Kumar et al. (2023), we make comparisons with our MAML + RGD approach as the proposed variant. We evaluate RGD not only based on the average performance across tasks but also on the Worst-K% of tasks in a fixed task pool. Our experiments on various benchmarks, including Omniglot 5-way 1-shot, Omniglot 20-way 1-shot, and *mini*ImageNet 5-way 1-shot, demonstrate significant improvements in the Worst-K% metric (Table 9). For example, on Omniglot 20-way 1-shot, our proposed reweighting scheme improves overall performance by **1.83%** and the Worst-10% performance by **2.28%**. Similarly, on the challenging *mini*ImageNet 5-way 1-shot benchmark, we achieve a substantial improvement of approximately **3%** across the board. Further results in this domain are discussed in Appendix D.7.

## 5 Conclusion, Limitations and Future Work

We introduced a re-weighted gradient descent (RGD) technique that effectively boosts the performance of deep learning across a wide range of tasks and domains. It is simple to implement and can be seamlessly integrated into existing algorithms with just two lines of code change. Our algorithm is derived from Kullback-Leibler distributionally robust optimization, a known method for improving model generalization.

While RGD shows promising results, it has the following limitations that warrant further investigation: (a) *outlier handling:* our current approach uses weight clipping to mitigate the impact of outliers. While empirically effective, a more principled approach to outlier robustness in DRO is worth exploring, and (b) *robustness to noise:* while RGD can handle benign noise, it can fail in the presence of adversarial/systematic noise. In the future, we plan to develop variants of RGD that can tolerate adversarial corruptions in the training data, while simultaneously improving the model generalization. Additionally, we plan to evaluate

our technique on large-scale tasks, such as fine-tuning Large Language Models (LLMs) and other foundation models. This will help us better understand the usefulness and limitations of our approach.

**Ethical Statement and Broader Impact**

Our proposed approach is compatible with any learning objective expressed as an expectation over samples. We showcased its effectiveness with various loss functions, including Mean Square Error, Cross Entropy, and others, outperforming previous state-of-the-art methods considerably. Implementing our approach is straightforward, and it has broad applicability across domains such as Natural Language Processing (NLP), Vision, and Time Series data. This paper presents work whose goal is to advance the field of Machine Learning. There are many potential societal consequences of our work, none which we feel must be specifically highlighted here.

**Acknowledgments**

We would like to thank Ahmad Beirami, Virginia Smith, Manzil Zaheer, Tian Li, and Maziar Sanjabi for providing detailed feedback on our paper. We would like to thank Elan Rosenfeld, and Tianbao Yang for pointing us to important prior works. We extend our sincere gratitude to Prateek Jain, Pradeep Shenoy, Anshul Nasery, Lovish Madaan, and the numerous dedicated members of the machine learning and optimization team at Google DeepMind India for their invaluable feedback and contributions to this work.

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

# Appendix

# Reproducibility Statement

Our proposed loss function is a single line of change. However, one would have to play around with the learning rate (generally lower than the baseline setting). Our experiments are based on public datasets and open-source code repositories. The proposed final formulation RGD requires **one line of code change**.

Suppose the per-sample loss is given. Example code for applying RGD in Jax is shown below.

```
import jax.numpy as jnp
import jax

def rgd_e(loss, temp=alpha, reduce=True):
    # alpha >0.
  out = loss * jnp.exp(
      jnp.clip(jax.lax.stop_gradient(loss), a_min=0, a_max=temp) / (temp + 1)
  )
  return out.sum() / len(out) if reduce else out
```

# A    Extended Related Work

## A.1    Per-Sample Reweighting

In this section, we review data reweighting techniques developed outside of the DRO community. The idea of re-weighting samples can be dated back to the works of Chawla et al. (2002); Zadrozny (2004), which pre-computed per-sample weights using certain prior knowledge. Recent approaches alleviate the need for human supervision by dynamically computing the per-sample weights. One of the early works in this category is AdaBoost, which is a popular boosting algorithm (Freund & Schapire, 1997). Similar to RGD, AdaBoost uses exponential weighting mechanism to reweight data points. However, AdaBoost is used for learning an ensemble of weak learners. Whereas, in this work, we are interested in learning a single model that can achieve better generalization guarantees. Furthermore, AdaBoost is only studied for supervised learning (in particular, classification and regression). In contrast, RGD can be applied on any learning task. Recent works of Leng et al. (2022); Lin et al. (2017) showed that certain modifications to standard cross entropy loss - that involve truncating its Taylor-series expansion - can improve the performance of DNNs. These techniques can be viewed as performing sample re-weighting. However, these techniques only apply to cross-entropy loss and are not easily extendable to general learning tasks.

Other approaches based on meta-learning have been proposed for class imbalance and label noise (Shu et al., 2019; Ren et al., 2018; Gonzalez & Miikkulainen, 2021). Many popular approaches in this line of work require training a separate neural network for re-weighting the data points (Ren et al., 2018; Shu et al., 2019). However, these approaches are seldom used in practice as the underlying bi-level optimization problem is hard to implement (Ivanova & Ablin, 2023). Unlike these approaches, our RGD algorithm does not require a separate neural network for re-weighting and thus doesn't add any computational overhead over vanilla training. Moreover, compared to existing sample re-weighting techniques, our approach applies to various learning tasks (see Section 4). Another line of work uses a history buffer which stores a snapshot of the trajectory of each point and facilitates giving more importance to points which leads to more learning in the model (Zhang & Pfister, 2021). Other approaches, such as Zhu et al. (2022b), use reinforcement learning to learn the per-sample weights using a "pretraining-boosting" two-stage MDP curriculum where the agent network is firstly pre-trained and optimized for deployment in the classification problem. Another line of work has considered sample re-weighting in the presence of outliers (Kumar et al., 2010; De La Torre & Black, 2003; Jiang et al., 2014a;b; Wang et al., 2017; Li et al., 2021; 2023). These works down-weight points with high-loss value. The rationality behind this lies in the idea that these high-loss samples are more likely to be outliers and, thus, should be ignored during the training process. Finally, works such Castells et al. (2020) propose a confidence-aware loss proportional to the lowest loss of that sample. They use a threshold ($\gamma$) to decide how practical or important each point is.

An emerging line of work on optimization focuses on designing sample re-weighting for improving the convergence speed of SGD (Katharopoulos & Fleuret, 2018; El Hanchi et al., 2022) by decreasing the variance in SGD iterates. Note that in contrast to these works which aim to minimize the ERM objective, we aim to solve the DRO objective which has better generalization guarantees than ERM in high variance, low sample complexity regime.

## A.2 Pre-conditioning

Pre-conditioning can usually mean normalization of inputs, batch normalization, or scaling gradients in a few directions. This section predominantly discusses techniques that focus on scaling gradients in a few directions. A common technique to improve the training speed in deep learning is using adaptive step-size optimization methods, such as the Newton method, which takes advantage of the second-order gradients. However, computing the Hessian matrix is computationally intensive, leading to Quasi-Newton methods: methods that approximate the value of the Hessian instead of computing them every time Le et al. (2011). Another popular alternative is to use an element-wise adaptive learning rate, which has shown great promise in deep learning. Some of the popular techniques here include ADAgrad (Duchi et al., 2011), RMSProp (Ruder, 2016), ADAdelta (Zeiler, 2012). For instance, ADAgrad is a diagonal pre-conditioning technique where the pre-conditioning across each dimension is computed as the inverse square root of the norms of gradients along that dimension accumulated over training. Unfortunately, this accumulation of gradients makes it susceptible to falling in a saddle point as the scaling factor decreases monotonically.

## A.3 Curriculum Learning

Another important research area that has explored data reweighting is Curriculum Learning (CL). CL, originally introduced by Bengio et al. (2009), is a vast domain focussing on how the model should be taught, and draws inspiration from how humans learn concepts. For instance, humans generally grasp on to easier concepts such as basic shapes (triangle, rectangle, etc.) before moving on to learning significantly more complex structures (heptagram, triquetra, etc.). Curriculum learning strategies have been widely used in various areas of machine learning and involves finding a way to rank samples, as well as the right pacing functions for introducing more difficult data in our training. The techniques developed for CL have typically focused on giving importance to easier samples at the beginning of training, and slowly progressing towards harder samples (Bengio et al., 2009; Chen & Gupta, 2015; Tudor Ionescu et al., 2016; Pentina et al., 2015; Shi et al., 2015; Spitkovsky et al., 2010; Zaremba & Sutskever, 2014). In contrast, DRO focuses the learning on harder samples throughout the training process. That being said, there have also been a class of works in CL which showed the learning harder examples first, and then moving to easier ones could lead to improved performance in certain conditions, through Hard Example mining (HEM) or anti-curriculum (Jesson et al., 2017; Shrivastava et al., 2016; Wang & Vasconcelos, 2018; Zhou et al., 2020; Braun et al., 2017; Pi et al., 2016). There have been predominantly three classes of CL, and various amalgamations of these in literature (Soviany et al., 2022).

*Vanilla CL:* The vanilla CL usually involves a pre-defined notion of hardness. For example, Bengio et al. (2009) used geometric shapes to clearly differentiate easy and hard samples. Others such as Spitkovsky et al. (2010) exploited the length of sequences as a signal for difficulty.

*Self-Paced Learning (SPCL):* This differs from the vanilla CL with respect to the evaluation of difficulty. This concept of "difficulty" is not known beforehand and is measured repeatedly during training. Works such as Kumar et al. (2010) used the likelihood of the prediction to rank the samples. Other works such as Lee & Grauman (2011) used objectness as a measure to define the training schedule.

*Balanced Curriculum (BCL):* In addition to prior works such as vanilla CL and SPCL, balanced curriculum approaches come with an added condition of diversity within a batch. These constraints (on classes, image regions, etc.) help the model learn robust features and not overfit to the spurious correlations of the easy samples (Zhang et al., 2015; Soviany, 2020).

In this work, we will resort to only describing few works which focus on instance level reweighting of data points (Kumar et al., 2010; Li et al., 2017; Kumar et al., 2011; Pi et al., 2016; Liang et al., 2016; Fan et al.,

2017; Li et al., 2017). Most of these works (Kumar et al., 2011; Fan et al., 2017) follow a binary weighting mechanism of $\{0, 1\}$ to decide whether the model should learn using the current sample or not. Others such as Li et al. (2017); Pi et al. (2016); Liang et al. (2016) are more continuous in the weighing mechanism and generally give higher weights to samples with lower losses. This makes them fundamentally different from RGD attempts to achieve in this work. RGD attempts to focus more heavily on the harder samples throughout training and does so using a simple closed form expression of the loss.

### A.4  Comparison with existing KL-DRO optimization approaches

In this section, we discuss a few more aspects of the related work that weren't discussed in the main paper. We specifically focus on prior works that developed algorithms for KL divergence based DRO. The earliest work on KL-DRO dates back to 2013 by Hu & Hong (2012). However, it was only recently that these works have become widespread in deep learning. The RECOVER algorithm by Qi et al. (2021) was one of the early works to scale KL-DRO to deep neural networks. It attempted to solve the non-convex DRO problem with a duality-free stochastic method by formulating the min-max formulation into an equivalent stochastic compositional problem. ABSGD (Qi et al., 2023b) and SCDRO (Qi et al., 2023a) improved upon RECOVER by designing more efficient algorithms. However, the performance of these algorithms on large-scale models and datasets is not rigorously studied, as the Imagenet-LT, iNaturalist experiments conducted in these works started from a pre-trained network and only finetuned the last layer. In contrast, in this work, we learn the entire ViT-S model from scratch and show improved generalization.

## B  Proofs of Section 3

### B.1  Proof of Proposition 3.1

**Proposition B.1.** *Consider DRO with KL-divergence-based uncertainty set. Assume that the data set $(z_i)_{i=1}^n$ is comprised of unique points (i.e, no repeated data points). Then $\min_{\theta \in \Theta} \widehat{R}_{D,n}$ can be rewritten as*

$$\min_{\theta \in \Theta} \frac{1}{\gamma} \log \mathbb{E}_{\widehat{P}_{data}}[e^{\gamma \ell(z;\theta)}],$$

*for some constant $\gamma > 0$ that is independent of $\theta$.*

*Proof.* Recall the empirical DRO risk $\widehat{R}_{D,n}(\theta)$ is defined as

$$\widehat{R}_{D,n}(\theta) \coloneqq \sup_{P' : D(P' || \widehat{P}_{\text{data}}) \leq \rho} \mathbb{E}_{P'}[\ell(z, \theta)]$$

Using Lagrangian duality, we rewrite $\widehat{R}_{D,n}(\theta)$ as

$$\sup_{P' : D(P' || \widehat{P}_{\text{data}}) \leq \rho} \mathbb{E}_{P'}[\ell(z, \theta)] = \sup_{P'} \inf_{\beta > 0} \mathbb{E}_{P'}[\ell(z, \theta)] - \beta(D(P' || \widehat{P}_{\text{data}}) - \rho)$$

$$\overset{(a)}{=} \sup_{P'} \inf_{\beta > 0} \mathbb{E}_{P'}[\ell(z, \theta)] - \beta(\mathbb{E}_{P'}[\log dP'] + \log n - \rho),$$

where $(a)$ follows from our choice of divergence, and the fact that the data points $\{z_i\}_{i=1}^n$ are all unique (i.e, no repetitions). Observe that the objective in the last expression is concave in $P'$ and linear in $\beta$. So the max-min problem above is concave-convex. Using Lagrangian duality to swap the order of min and max, we get

$$\sup_{P' : D(P' || \widehat{P}_{\text{data}}) \leq \rho} \mathbb{E}_{P'}[\ell(z, \theta)] = \inf_{\beta > 0} \sup_{P'} \mathbb{E}_{P'}[\ell(z, \theta)] - \beta(\mathbb{E}_{P'}[\log dP'] + \log n - \rho).$$

This shows that minimizing $\widehat{R}_{D,n}$ is equivalent to the following problem

$$\inf_{\beta > 0, \theta \in \Theta} \sup_{P'} \mathbb{E}_{P'}[\ell(z, \theta)] - \beta(\mathbb{E}_{P'}[\log dP'] + \log n - \rho).$$

For any fixed $\beta, \theta$, the inner supremum is attained at a $P'$ that satisfies (see Theorem 1 of Hsieh et al. (2019))

$$P'(z) \propto \exp\left(\ell(z,\theta)/\beta\right).$$

This can be derived using the following first order optimality condition: $\forall z, \ell(z,\theta) - \beta \log P'(z) - \beta = c$ for some constant $c$. Substituting this in the previous equation, we get the following equivalent optimization problem

$$\inf_{\beta>0} \inf_{\theta \in \Theta} \beta \log \mathbb{E}_{P'}[e^{\ell(z,\theta)/\beta}] - \beta(\log n - \rho).$$

Letting $\gamma^{-1}$ be the minimizer of the outer minimization problem, we get the required result. $\qquad\square$

## B.2 Proof of Proposition 3.2

*Proof.* Note that whenever $\ell(z;)$ is convex, so is $f(\theta) := \mathbb{E}_{z \sim \widehat{P}_{\text{data}}} \exp(\gamma\ell(z;))$. It is easy to check that the function $f$ and the constraint set $\Theta$ satisfy the conditions in Shamir & Zhang (2013, Theorem 2). From this, we conclude that:

$$\mathbb{E}_{\theta_T} f(\theta_T) - \inf_{\theta \in \Theta} f(\theta) = O\left(\frac{\log T}{\sqrt{T}}\right) \tag{3}$$

The proof of equation 3 for the step size sequence $\eta_t = \frac{C}{\sqrt{t}}$ follows from the statement of Shamir & Zhang (2013, Theorem 2). The case of the constant step-size $\eta_t = \frac{C}{\sqrt{T}}$ follows by a simple modification of the proof of Shamir & Zhang (2013, Theorem 2) where we substitute the appearance of $1/\sqrt{t}$ due to the step size with $1/\sqrt{T}$. We now convert the guarantees in Equation 3 to guarantees in terms of $\log f(\theta)$. Let $\theta^* \in \arg\inf_{\theta \in \Theta} f(\theta)$. By our assuumption, $\ell(z;)$ is bounded above and below. So $\log f(\theta_T) - \log f(\theta^*) \leq \bar{C}(f(\theta_T) - f(\theta^*))$ for some $\bar{C}$. Combining this with equation 3, we conclude the statement of the proposition. $\qquad\square$

## B.3 Other Divergences

$\chi^2$**-divergence.** Consider $\chi^2$-divergence which is defined as

$$D(P'||P) = \mathbb{E}_P\left[\left(\frac{dP'}{dP} - 1\right)^2\right].$$

We now follow a similar argument as in the proof of Proposition 3.1 to derive an equivalent expression for the DRO objective. We have

$$\sup_{P':D(P'||\widehat{P}_{\text{data}})\leq\rho} \mathbb{E}_{P'}[\ell(z,\theta)] = \sup_{P'} \inf_{\beta>0} \mathbb{E}_{P'}[\ell(z,\theta)] - \beta(D(P'||\widehat{P}_{\text{data}}) - \rho)$$

$$\stackrel{(a)}{=} \sup_{P'} \inf_{\beta>0} \mathbb{E}_{P'}[\ell(z,\theta)] - \beta(\mathbb{E}_{P'}[dP'/d\widehat{P}_{\text{data}}] - 1 - \rho)$$

$$\stackrel{(b)}{=} \sup_{P'} \inf_{\beta>0} \mathbb{E}_{P'}[\ell(z,\theta)] - \beta(n\mathbb{E}_{P'}[dP'] - 1 - \rho)$$

$$\stackrel{(c)}{=} \inf_{\beta>0} \sup_{P'} \mathbb{E}_{P'}[\ell(z,\theta)] - \beta(n\mathbb{E}_{P'}[dP'] - 1 - \rho),$$

where $(a), (b)$ follow from the definition of the divergence and $(c)$ follows from Lagrangian duality. Now, consider the DRO optimization problem

$$\sup_{P':D(P'||\widehat{P}_{\text{data}})\leq\rho} \mathbb{E}_{P'}[\ell(z,\theta)] = \inf_{\beta>0,\theta\in\Theta} \sup_{P'} \mathbb{E}_{P'}[\ell(z,\theta)] - \beta(n\mathbb{E}_{P'}[dP'] - 1 - \rho).$$

Suppose the loss $\ell$ is positive. For any fixed $\theta, \beta$, the inner supremum in the above optimization problem is attained at a $P'$ that satisfies

$$P'(z) \propto \left(\ell(z,\theta)/\beta n + 1\right).$$

This follows from the first order optimality conditions. This gives rise to the re-weighting scheme $g(x) = x + \tau$, for some appropriately chosen $\tau$.

**Reverse KL divergence.** The reverse KL-divergence is defined as

$$D(P'||P) = \mathbb{E}_P\left[-\log \frac{dP'}{dP}\right].$$

Using similar arguments as above, we can rewrite the DRO optimization problem as

$$\sup_{P':D(P'||\widehat{P}_{\text{data}})\leq\rho} \mathbb{E}_{P'}[\ell(z,\theta)] = \inf_{\beta>0,\theta\in\Theta} \sup_{P'} \mathbb{E}_{P'}[\ell(z,\theta)] + \beta(\mathbb{E}_P[\log dP'] - \log n - \rho).$$

For any fixed $\theta, \beta$, the inner supremum in the above optimization problem is attained at a $P'$ that satisfies

$$P'(z) \propto \frac{1}{\tau - \ell(z,\theta)},$$

for some appropriate $\tau$. This gives rise to the re-weighting scheme $g(x) = 1/(\tau - x)$. We call this algorithm RGD-REVKL. In practice, we modify this re-weighting function

It can be implemented using the following pseudocode in Jax.

```
import jax.numpy as jnp
import jax

def rgd_t(loss, temp=alpha, reduce=True):
    # alpha >0.
  out = loss * (1 -
      jnp.clip(jax.lax.stop_gradient(loss), a_min=0, a_max=temp) / (temp + 1))** (-1)
  return out.sum() / len(out) if reduce else out
```

## C  Choice of divergence in RGD

RGD-REVKL is a more aggressive weighing scheme in comparison to RGD. This is fairly simple to show if you re-write both the reweighting techniques using Taylor series expansion. RGD-REVKL multiplies the loss $l$ with $(1 + l + l^2 + \dots)$. Whereas, RGD multiplies the loss $l$ with $(1 + l + l^2/2! + l^3/3! + \dots)$. RGD-REVKL is a more aggressive weighing scheme than RGD, and the choice between the two schemes should depend on the problem. Some preliminary results on the class imbalance setting is depicted in Table 10. For RGD-$\chi^2$ $g(u) = u + \tau$, we also clip $u$ to be $\min(u,\tau)$. Similarly, for RGD-REVKL, $g(u) = \frac{1}{\tau-u}$, we set $\tau = 1$ and clipped $u$ as $\min(u,t)/(t+1)$ where for all practical purposes - $t$ has a similar grid search and function as $\tau$ from other divergences.

Our search space and clipping involved the same grid search space as our RGD algorithm as described in the reproducibility statement.

Table 10: Test Accuracy of ResNet-32 on Long-Tailed CIFAR-10, and CIFAR-100 dataset.

| Dataset | CIFAR-10 | | | | | | | CIFAR-100 | | | | | | |
|---|---|---|---|---|---|---|---|---|---|---|---|---|---|---|
| Loss / Imbalance Factor | 200 | 100 | 50 | 20 | 10 | 1 | **Avg.** | 200 | 100 | 50 | 20 | 10 | 1 | **Avg.** |
| Cross Entropy (CE) | | | | | | | | | | | | | | |
| Default | 65.98 | 70.36 | 74.81 | 82.23 | 86.39 | 92.89 | 78.78 | 34.84 | 38.32 | 43.85 | 51.14 | 55.71 | 70.50 | 49.06 |
| RGD-REVKL (**Ours**) | 64.16 | 72.56 | 77.86 | 83.88 | 86.84 | 92.99 | 79.72 | 36.22 | 39.87 | 43.74 | 51.86 | 56.9 | 70.80 | 49.90 |
| RGD-$\chi^2$ (**Ours**) | 67.16 | 72.20 | 77.93 | 84.7 | 86.90 | 93.00 | 80.32 | 35.96 | 39.70 | 43.88 | 51.29 | 56.92 | 70.73 | 49.75 |
| RGD (**Ours**) | **67.90** | **73.75** | **79.63** | **85.44** | **88.00** | **93.27** | **81.33** | **38.62** | **41.89** | **46.40** | **53.48** | **58.5** | **71.30** | **51.70** |

## D  Additional Experimental Results and Missing Details

This section provides additional experimental results and details that are missing in the main paper. The search space and hyperparameters tuned in our experiments of are depicted in Appendix D.1

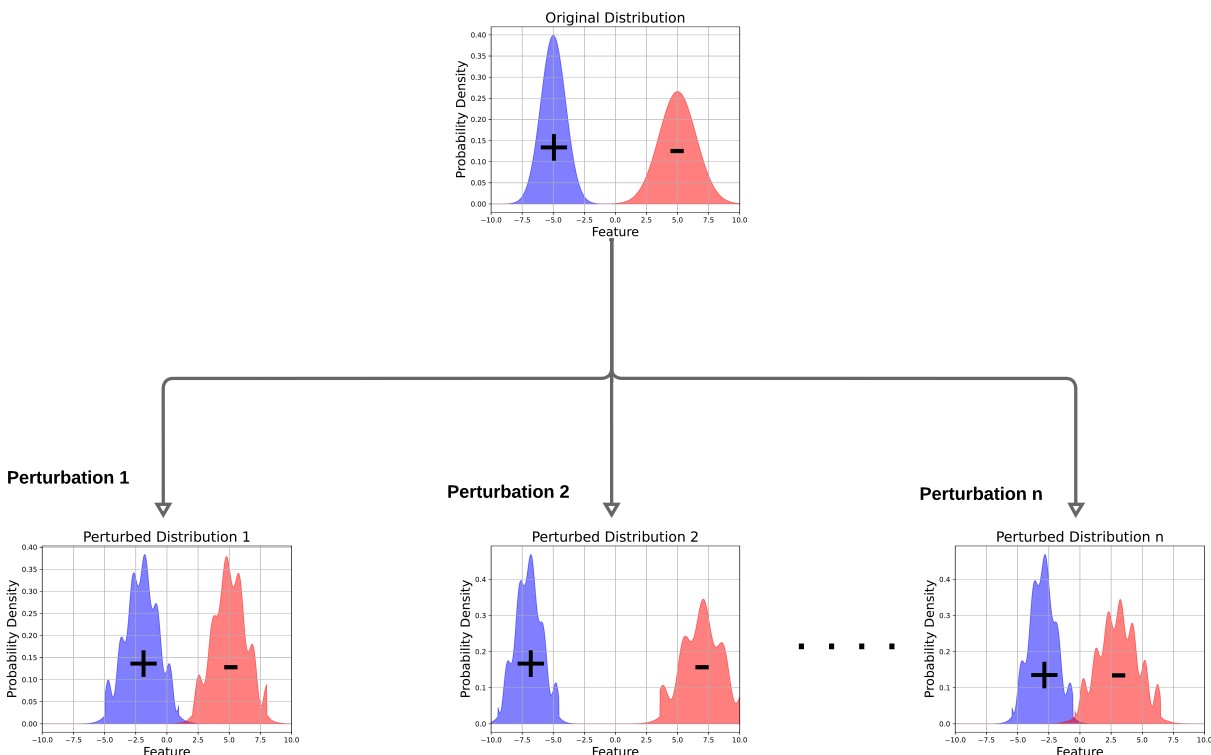

Figure 2: Figure illustrating Distributionally Robust Optimization (DRO). In contrast to ERM which learns a model that minimizes expected loss over original data distribution, DRO learns a model that performs well simultaneously on several perturbed versions of the original data distribution.

## D.1 Hyperparameter Tuning

In this section, we describe the common hyperparameter tuning space used across all experiments in our paper unless otherwise mentioned. The two parameters we tune were $\tau$ and `lr`. We use a simple grid search for $\tau$ in the order of $[1, 3, 5, 7, 9]$ across the experiments where the scaling factor ($\gamma$) is by default set as $\frac{1}{\tau+1}$. This allowed our loss to be bounded between 0,1 and helped fairly compare RGD-$\chi^2$ and RGD. The `lr` was tuned by a proxy of `lr_mult` where we scaled the learning rate by a fraction in the range $[0.5, 1.5]$. The effect of these hyperparameters ($\tau, \gamma$) is further depicted in Section 4.2.

### D.1.1 Existing KL-DRO techniques

For TERM, we use the batch (non-hierarchical) version (as shown in Algorithm 1 of Li et al. (2023)) - requiring two degrees of hyperparameters (tilting coefficient $t$ and the learning rate ($lr$)). For the tilting coefficient, we use a search space of Li et al. (2021): $\{0.2, 0.5, 1, 3, 5\}$. For the learning rate, we use a lr multiplier to the baseline run as $\{0.7, 0.8, 0.9, 1, 1.1, 1.2, 1.3\}$. For the stochastic version of TERM, which is identical to ABSGD (Qi et al., 2023b) baseline - requires an additional coefficient of moving average ($\beta$). We use a grid search space of $\{0.25, 0.5, 0.75\}$ for tuning $\beta$, and tune the learning rate in a similar fashion to TERM. We also tune $\lambda$ (similar to the tilting coefficient of TERM) in the search space $\{1, 3, 5, 7\}$.

## D.2 Toy Experiments

In this section, we perform a simple experiment to better understand the robustness properties of our proposed approach.

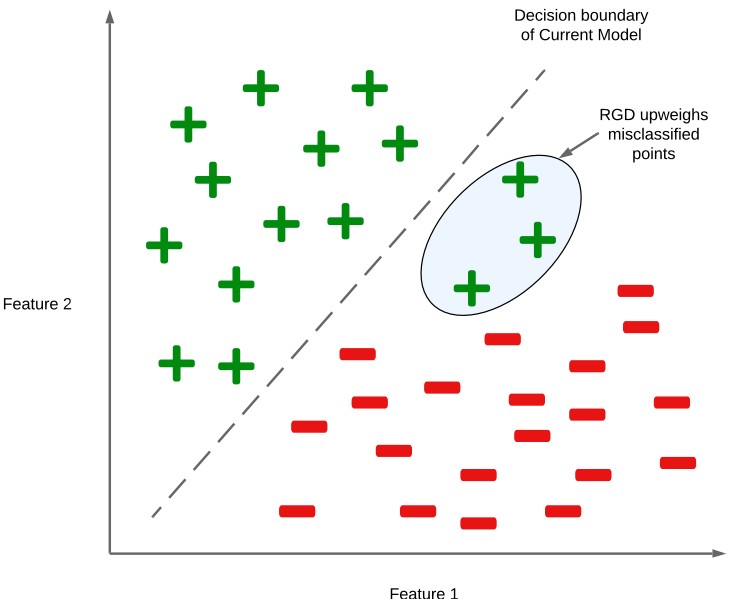

Figure 3: Figure illustrating the intuitive idea behind the working of RGD in the binary classification setting. RGD upweights the points which have high losses - points which have been misclassified by the model.

**Linear Regression with rare features:** We consider a linear regression problem where the covariates $x$ are sampled from the set $\{h(0), \ldots h(9)\}$. Here $h(\cdot)$ is a one-hot encoder that maps its inputs to a 10-dimensional vector. The label $y$ is generated according to the following linear model: $y = x^T \theta^*$, where $\theta^* \in \mathbb{R}^{10}$ is the regression vector which is sampled from a standard normal distribution. We construct an imbalanced dataset of tuples $\{(x_i, y_i)\}_{i=1}^n$ by taking 50 occurrences of the covariates $\{h(0) \ldots h(4)\}$ and only one occurrence of the remaining five covariates. We consider two algorithms for learning the unknown parameter vector $\theta^*$: (i) SGD on the mean squared error in predictions (MSE) (ii) RGD on the MSE loss. We set the step size to be 4 for both algorithms and plot the evolution of MSE and the Euclidean distance between the iterates ($\theta$) and the true parameter ($\theta^*$) (Figure 4). It can be seen that our method achieves better performance due to prioritization of samples with higher loss, which corresponds to rare directions in the dataset.

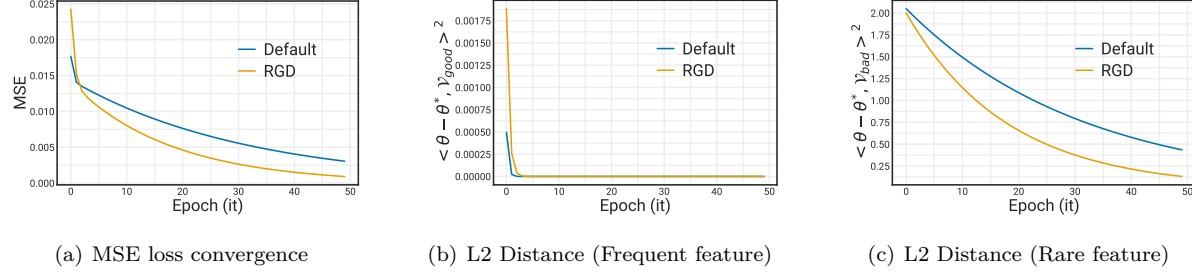

(a) MSE loss convergence      (b) L2 Distance (Frequent feature)      (c) L2 Distance (Rare feature)

Figure 4: Figure 4(a) showing the convergence of SGD, RGD algorithms for estimating the linear regression parameter. The L2 distance between the iterates ($\theta$) and the true parameter $\theta^*$ is studied in Figures 4(b) and 4(c). Specifically, Figure 4(b) depicts the squared error in the frequently appearing directions, where all the techniques perform equally well. However, when it comes to learning rare directions, our proposed approach is much better (Figure 4(c)).

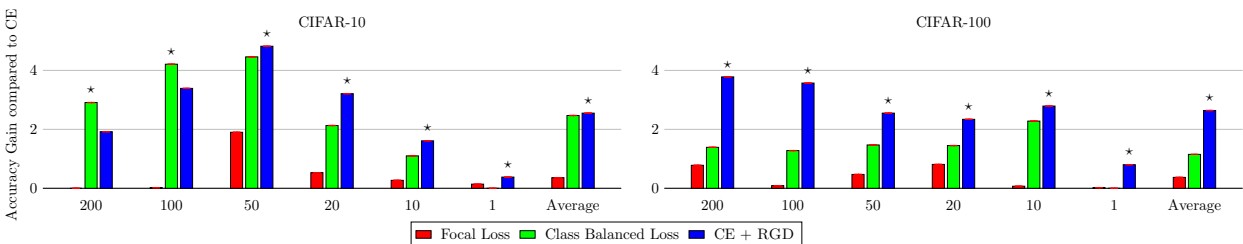

Figure 5: Experiment comparing RGD with baseline cross entropy Loss (CE), focal loss and class-balanced loss using a ResNet-32 backbone. $x$-axis represents the imbalance factor in the dataset.

## D.3  Comparisons against Sharpness Aware Minimization

Below we compare with Sharpness-Aware Minimization (SAM) (Foret et al., 2021) both from theoretical and empirical perspectives. From a theoretical perspective, SAM performs robust optimization in the weight space (that is SAM tries to learn a model that is robust to perturbations of weights). In contrast, RGD performs robust optimization in the distribution space. So, RGD and SAM are orthogonal to each other and can potentially be merged together to boost the performance. In Table 11, we compare performance of various approaches for CIFAR-10 optimization. It can be seen that RGD marginally boosts the performance of SAM. Although, more thorough experiments are needed to understand the utility of RGD on top of SAM.

Table 11: Comparison of RGD with SAM. Accuracy performance numbers have been reported over an average of 5 seeds.

| **Algorithm** | SGD | SAM | SAM + RGD |
|---|---|---|---|
| Accuracy | $96.80 \pm 0.04$ | $97.31 \pm 0.05$ | $\mathbf{97.41} \pm 0.09$ |

## D.4  Class Imbalance Experiments

This section briefly discusses additional results from our experiments on the Class Imbalance domain with imbalanced CIFAR-10 and CIFAR-100 datasets. It is well known that DRO outputs models with good tail performance (Duchi & Namkoong, 2018). Since RGD directly solves the DRO objective, our models are also naturally endowed with this property. To demonstrate this, we extend our experiments on linear regression to a more realistic image dataset, where some classes appear very rarely in the data set while some appear very frequently. We use the Long-Tailed CIFAR dataset, where we reduce the number of training samples per class according to an exponential function as proposed by Cui et al. (2019). We define the imbalance factor of a dataset as the number of training samples in the largest class divided by the smallest. Similar to the works of Shu et al. (2019), we use a ResNet-32 architecture for training. Apart from Cross Entropy loss, we also include Focal Loss (Lin et al., 2017) and Class Balanced Loss (Cui et al., 2019) as additional baselines. We also experimented with the long-tailed CIFAR-100 dataset and showed that our proposed approach could again show significant improvements. Figure 5 illustrates the performance of our approach in comparison to other state-of-the-art methods. Overall, in comparison to the SOTA approach in this task (Class Balanced Loss), our proposed approach brings about an improvement of **+0.79%**. A more comprehensive comparison with additional state-of-the-art baselines such as L2RW (Ren et al., 2018), and Meta-Weight-Net (Shu et al., 2019) is illustrated in Table 13. Although these models use additional data as a meta-validation-set, our proposed approach outperforms L2RW and is roughly competitive with the Meta-Weight-Net model.

Table 12 depicts the accuracy metric of models on various levels of the imbalance factor. From Table 12, we show that our proposed approach RGD outperforms other baselines such as Focal Loss and Class Balanced Loss by **+0.79%**. Furthermore, when models are trained on additional data, either by fine-tuning or by using a meta-learning framework to learn weights (such as Meta-Weight-Net and L2RW), we show that our proposed approach is competitively similar (**-0.22%**). Table 13 illustrates this analysis further. The

performance metrics of the baseline approaches were taken from Shu et al. (2019). Additional comparisons against other losses, such as cross-entropy with label smoothing and large margin softmax loss, are shown in Table 14.

For prior KL-DRO benchmarks such as TERM, we perform a grid-search where we tune the tilting coefficient ($t$) in the space $\{0.1, 0.3, 0.5, 0.7, 1, 2, 5\}$. Furthermore, we also tune the learning rate in the space $[5e - 3, 1]$. Furthermore, for ABSGD, we replicate the baseline numbers from their paper which performs the same setup of experiments as us.

Table 12: Test Accuracy of ResNet-32 on Long-Tailed CIFAR-10, and CIFAR-100 dataset.

| Dataset | CIFAR-10 | | | | | | | CIFAR-100 | | | | | | |
|---|---|---|---|---|---|---|---|---|---|---|---|---|---|---|
| Loss / Imbalance Factor | 200 | 100 | 50 | 20 | 10 | 1 | **Avg.** | 200 | 100 | 50 | 20 | 10 | 1 | **Avg.** |
| Focal Loss Lin et al. (2017) | 65.29 | 70.38 | 76.71 | 82.76 | 86.66 | 93.03 | 79.14 | 35.62 | 38.41 | 44.32 | 51.95 | 55.78 | 70.52 | 49.43 |
| Class Balanced Loss Cui et al. (2019) | **68.89** | **74.57** | 79.27 | 84.36 | 87.49 | 92.89 | 81.25 | 36.23 | 39.60 | 45.32 | 52.59 | 57.99 | 70.50 | 50.21 |
| Cross Entropy (CE) | | | | | | | | | | | | | | |
| Default | 65.98 | 70.36 | 74.81 | 82.23 | 86.39 | 92.89 | 78.78 | 34.84 | 38.32 | 43.85 | 51.14 | 55.71 | 70.50 | 49.06 |
| RGD-RevKL (**Ours**) | 64.16 | 72.56 | 77.86 | 83.88 | 86.84 | 92.99 | 79.72 | 36.22 | 39.87 | 43.74 | 51.86 | 56.9 | 70.80 | 49.90 |
| RGD (**Ours**) | 67.90 | 73.75 | **79.63** | **85.44** | **88.00** | **93.27** | **81.33** | **38.62** | **41.89** | **46.40** | **53.48** | **58.5** | **71.30** | **51.70** |

Table 13: Test Accuracy of ResNet-32 on Long-Tailed CIFAR-10, and CIFAR-100 dataset. We use the symbol $\star$ to denote approaches that use additional data (as the meta-dataset). We use *underline* symbol to depict performances which are second-best across baselines. Our experiments show that we can get competitively similar performance to such models as well without training a second neural network.

| Dataset | CIFAR-10 | | | | | | | CIFAR-100 | | | | | | |
|---|---|---|---|---|---|---|---|---|---|---|---|---|---|---|
| Loss / Imbalance Factor | 200 | 100 | 50 | 20 | 10 | 1 | **Avg.** | 200 | 100 | 50 | 20 | 10 | 1 | **Avg.** |
| Fine-tuning $\star$ | 66.08 | 71.33 | 77.42 | 83.37 | 86.42 | _93.23_ | 79.64 | _38.22_ | 41.83 | 46.40 | 52.11 | 57.44 | 70.72 | 51.12 |
| L2RW Ren et al. (2018) $\star$ | 66.51 | _74.16_ | 78.93 | 82.12 | 85.19 | 89.25 | 77.69 | 33.38 | 40.23 | 44.44 | 51.64 | 53.73 | 64.11 | 47.92 |
| Meta-Weight-Net Shu et al. (2019) $\star$ | **68.91** | **75.21** | **80.06** | _84.94_ | _87.84_ | 92.66 | **81.60** | 37.91 | **42.09** | **46.74** | **54.37** | _58.46_ | _70.37_ | _51.65_ |
| Cross Entropy (CE) | | | | | | | | | | | | | | |
| Default | 65.98 | 70.36 | 74.81 | 82.23 | 86.39 | 92.89 | 78.78 | 34.84 | 38.32 | 43.85 | 51.14 | 55.71 | 70.50 | 49.06 |
| RGD-RevKL (**Ours**) | 64.16 | 72.56 | 77.86 | 83.88 | 86.84 | 92.99 | 79.72 | 36.22 | 39.87 | 43.74 | 51.86 | 56.9 | 70.80 | 49.90 |
| RGD (**Ours**) | _67.90_ | 73.75 | _79.63_ | **85.44** | **88.00** | **93.27** | _81.33_ | **38.62** | _41.89_ | _46.40_ | _53.48_ | **58.5** | **71.30** | **51.70** |

Table 14: Additional Ablation study comparing with Label smoothing and Large Margin Softmax Loss. Test Accuracy of ResNet-32 on Long-Tailed CIFAR-10, and CIFAR-100 dataset.

| Dataset | CIFAR-10 | | | | | | | CIFAR-100 | | | | | | |
|---|---|---|---|---|---|---|---|---|---|---|---|---|---|---|
| Loss / Imbalance Factor | 200 | 100 | 50 | 20 | 10 | 1 | **Avg.** | 200 | 100 | 50 | 20 | 10 | 1 | **Avg.** |
| Cross Entropy (CE) | | | | | | | | | | | | | | |
| + Label Smoothing | 61.22 | 73.80 | 77.95 | 84.40 | 86.96 | 92.18 | 79.42 | 37.14 | 41.05 | 44.76 | 50.67 | 57.74 | 70.97 | 50.39 |
| + Large Margin Softmax (LMS) | **68.67** | 72.78 | 78.84 | 85.23 | **88.26** | 92.75 | 81.09 | 36.77 | 40.38 | 45.24 | 51.25 | 56.9 | 71.01 | 50.26 |
| RGD (**Ours**) | 67.90 | **73.75** | **79.63** | **85.44** | 88.00 | **93.27** | **81.33** | **38.62** | **41.89** | **46.40** | **53.48** | **58.5** | **71.30** | **51.70** |

## D.5 Vanilla Classification

This section briefly discusses a few additional results from our experiments on standard supervised learning (in particular vanilla classification). Table 15 depicts the performance of our other variant RGD-RevKL in comparison to the baseline approach.

**EfficientNet finetuning.** We also show fine-tuning improvements of EfficientNet-v2-l over various tasks such as Cars and Food101 as depicted in Table 16. In these experiments, we take a pre-trained EfficientNet backbone and fine-tune it for various tasks.

**DieT-S for ImageNet-1K classification.** Similarly, we also report that RGD boosts the performance of the baseline DeiT-S model by 0.1% when trained from scratch on the Imagenet-1K benchmark as depicted

in Table 16. Note that similar to our setup in ViT-S on Imagenet-1K benchmark, we perform no tuning, and simply use $\tau = 1$, and same learning rate as baseline.

**MLP for classification.** We also demonstrate that our proposed approach is simple and shows significant improvements, not only for SOTA approaches but also basic MLP procedures as depicted in Table 18, Table 19, and Table 20. These tables help showcase that the simple addition of our proposed approach does show significant improvements of **+2.77%** (in accuracy) on multi-class and **+1.77%** (in AUROC) on binary class tasks respectively.

Table 15: Additional Ablation study to showcase the gain achieved by using RGD for various tasks in GLUE benchmark for bert-base-uncased.

| bert-base-uncased | MNLI | QQP | QNLI | SST-2 | MRPC | RTE | COLA | Avg |
|---|---|---|---|---|---|---|---|---|
| Default | 81.33 | 89.62 | 87.93 | 90.63 | **89.55** | 67.19 | 54.53 | 80.11 |
| RGD-RevKL (**Ours**) | 82.97 | 89.87 | **90.79** | 91.28 | 88.54 | 71.23 | 59.28 | 81.97 |
| RGD (**Ours**) | **83.06** | **91.06** | 90.35 | **91.78** | 88.28 | **71.48** | **58.56** | **82.05** |

Table 16: Ablation study to showcase the gain achieved by using RGD for various tasks in Vision benchmarks for ViT-S on Imagenet-1K, DeiT-S on Imagenet-1K and EfficientNet-v2-l on Finetuning tasks.

| | Imagenet-1K (ViT-S) | Imagenet-1K (DeiT-S) | Cars-FineTuning | Food101-FineTuning |
|---|---|---|---|---|
| Default | 78.1 | 80.05 | 92.03 | 92.65 |
| RGD (**Ours**) | **79.0** | **80.13** | **92.62** | **92.75** |

**MiniGPT Pre-training.** We extend our work to large-scale tasks in NLP such as LLM pre-training which has become more prevalent over the recent years. Our experiments on miniGPT pre-training, as illustrated in this section, showcase the efficacy of our approach in these settings. miniGPT (Zhu et al., 2024) is a minimal implementation of a decoder-only transformer language model. We consider a 6 layer model and train on the lm1B small dataset which has 1B tokens. We trained for 100K steps with a batch size of 256. We used the default learning rate of 0.0016 for the baseline. For RGD, we fix the clipping threshold to 1 and tune the learning rate. We achieved 1% improvement on the eval log-perplexity score. Table 17 illustrates our results in this setting.

### D.6 Tabular Classification

This section discusses a few additional results from our experiments on Tabular classification. Table 21 depicts our proposed approach's accuracy compared to other baselines on multi-class tabular datasets. Our method outperforms previous SOTA in this problem by **+1.27%**. Furthermore, Table 22 illustrates the AUROC score of our proposed approach in comparison to state-of-the-art baselines on binary-class tabular datasets. Our approach shows an improvement of **+1.5%** in this setting as well. The performance metrics of the baseline approaches were taken from Majmundar et al. (2022).

Table 17: Results on standard Lm1b pre-training task with miniGPT. Off-the-hat addition of RGD leads to improvements of evaluation log-perplexity by 1%.

| Dataset | Lm1b eval Perplexity |
|---|---|
| Default | 2.9218 |
| RGD (**Ours**) | **2.8979** |

Table 18: Results on standard multi-class tabular datasets (Accuracy): The bottom partition shows results of our method with RGD loss. We show that the addition of our proposed approach significantly outperforms the base MLP model.

| Algorithm | FMNIST | CIFAR10 | MNIST | CovType | Avg. |
|---|---|---|---|---|---|
| Default Majmundar et al. (2022) | 87.62 | 16.50 | 96.95 | 65.47 | 66.64 |
| RGD-RevKL (**Ours**) | 88.52 | 20.31 | **97.86** | 68.81 | 68.875 |
| RGD (**Ours**) | **89.03** | **21.32** | 97.57 | **69.73** | **69.41** |

Table 19: Results on standard binary-class tabular datasets (Accuracy): The bottom partition shows results of our method with RGD loss. We show that the addition of our proposed approach significantly outperforms the base MLP model.

| Algorithm | Obesity | Income | Criteo | Thyroid | Avg. |
|---|---|---|---|---|---|
| Default | 58.1 | 84.36 | 74.28 | 50 | 66.69 |
| RGD-RevKL (**Ours**) | 59.12 | 84.5 | 75.06 | **57.3** | 69 |
| RGD (**Ours**) | **58.83** | **85.5** | **75.87** | 56.04 | **69.1** |

Table 20: Results on standard binary-class tabular datasets (AUROC): The bottom partition shows results of our method with RGD loss. We show that the addition of our proposed approach significantly outperforms the base MLP model.

| Algorithm | Obesity | Income | Criteo | Thyroid | Avg. |
|---|---|---|---|---|---|
| MLP | 52.3 | 89.39 | 79.82 | 62.3 | 70.95 |
| RGD-RevKL (**Ours**) | 54.31 | **91.1** | 79.85 | 62.25 | 71.88 |
| RGD (**Ours**) | **55.96** | 90.8 | **80.1** | **64** | **72.72** |

Table 21: Results on standard multi-class tabular datasets (Accuracy): The bottom partition shows results of our method with RGD loss. We show that the addition of our proposed approach significantly outperforms existing methods, as well as SOTA.

| Algorithm | FMNIST | CIFAR10 | MNIST | CovType | Avg. |
|---|---|---|---|---|---|
| MLP | 87.62 | 16.50 | 96.95 | 65.47 | 66.64 |
| RF Breiman (2001) | 88.43 | 42.73 | 97.62 | 71.37 | 75.04 |
| GBDT Friedman (2001) | 88.71 | 45.7 | **100** | 72.96 | 76.84 |
| RF-G Rahimi & Recht (2008) | 89.84 | 29.32 | 97.65 | 71.57 | 72.10 |
| MET-R Majmundar et al. (2022) | 88.84 | 28.94 | 97.44 | 69.68 | 71.23 |
| VIME Yoon et al. (2020) | 80.36 | 34.00 | 95.77 | 62.80 | 68.23 |
| DACL+ Verma et al. (2021) | 81.40 | 39.70 | 91.40 | 64.23 | 69.18 |
| SubTab Ucar et al. (2021) | 87.59 | 39.34 | 98.31 | 42.36 | 66.90 |
| TabNet Arik & Pfister (2021) | 88.18 | 33.75 | 96.63 | 65.13 | 70.92 |
| MET Majmundar et al. (2022) | **91.68** | 47.82 | 99.19 | 76.71 | 78.85 |
| MET-S | | | | | |
| Default Majmundar et al. (2022) | 90.94 | 48.00 | 99.01 | 74.11 | 78.02 |
| RGD-RevKL (**Ours**) | 91.12 | 49.17 | 99.28 | 79.41 | 79.75 |
| RGD (**Ours**) | 91.54 | **49.54** | 99.69 | **79.72** | **80.12** |

Table 22: Results on standard binary-class tabular datasets (AUROC): The bottom partition shows results of our method with RGD loss. We show that the addition of our proposed approach significantly outperforms existing methods, as well as SOTA.

| Algorithm | Obesity | Income | Criteo | Thyroid | Avg. |
|---|---|---|---|---|---|
| MLP | 52.3 | 89.39 | 79.82 | 62.3 | 70.95 |
| RF Breiman (2001) | 64.36 | 91.53 | 77.57 | 99.62 | 83.27 |
| GBDT Friedman (2001) | 64.4 | 92.5 | 78.77 | 99.34 | 83.75 |
| RF-G Rahimi & Recht (2008) | 54.45 | 90.09 | 80.32 | 52.65 | 69.37 |
| MET-R Majmundar et al. (2022) | 53.2 | 83.54 | 79.17 | 82.03 | 74.49 |
| VIME Yoon et al. (2020) | 57.27 | 87.37 | 74.28 | 94.87 | 78.45 |
| DACL+ Verma et al. (2021) | 61.18 | 89.01 | 75.32 | 86.63 | 78.04 |
| SubTab Ucar et al. (2021) | 64.92 | 88.95 | 76.57 | 88.93 | 79.00 |
| TabNet Arik & Pfister (2021) | 69.40 | 77.30 | 80.91 | 96.98 | 81.15 |
| MET-S | | | | | |
| Default Majmundar et al. (2022) | 71.84 | 93.85 | 86.17 | 99.81 | 87.92 |
| RGD-REVKL **(Ours)** | 76.23 | 93.90 | 86.92 | 99.82 | 89.22 |
| RGD **(Ours)** | **76.87** | **93.96** | **86.98** | **99.92** | **89.43** |

### D.7 Meta-Learning

This section discusses some additional results from our experiments in the meta-learning domain. Table 23 depicts a complete table and comparison of our proposed approach on the MAML baseline compared to others. Overall, we notice improvement across the board, especially in the outliers, as shown in the Worst-k% metrics. Note that although our RGD has been applied on MAML (Finn et al., 2017) in our current experiments, our approach is analogous to the model and can be extended to other meta-learning techniques such as Protonet (Snell et al., 2017), CNAPs (Requeima et al., 2019), etc. as well. The performance metrics of the baseline approaches were taken from (Kumar et al., 2023).

### D.8 DomainBed

#### D.8.1 DomainBed Benchmark

In this section, we describe the DomainBed benchmark, a challenging benchmark used to study the out-of-domain generalization capabilities of our model. To briefly explain, consider the dataset `PACS`, which consists of Photos, Art, cartoons, and sketches of the same set of classes (for instance, dogs and cats, amongst others). The goal of the task is to learn from three of these domains and evaluate the performance of the left-out domain (similar to a k-fold cross-validation). By doing so, we can assess the out-of-domain generalization performance of our models. In general, the metric used in this domain involves taking an average of the performance of the different k-fold splits. More information about this benchmark is available at Gulrajani & Lopez-Paz (2021).

#### D.8.2 Additional Results

In this section, we briefly discuss additional results from our DomainBed experiments. Table 24 depicts a complete table and comparison of our proposed approach to a multitude of state-of-the-art approaches in this field. Furthermore, we also show that our proposed approach outperforms previous SOTA by **+0.7%**.

Moreover, we also report the performance improvements when RGD is trained with model weight averaging methods such as SWAD Cha et al. (2021). Table 29 depicts the performance improvements of RGD over SWAD.

Furthermore, we also present the per-environment breakdown of our approach in various datasets in Table 25, Table 26, Table 27, and Table 28 for PACS, VLCS, OfficeHome, and DomainNet respectively. The performance metrics of the baseline approaches were taken from Gulrajani & Lopez-Paz (2021).

Table 23: Results on meta-learning datasets. We report the Worst-K% performance as well to help study the performance distribution over all tasks. Overall, we expect our reweighting scheme to give more importance to those tasks which are difficult and rare. We show that the addition of our proposed approach significantly outperforms existing methods as shown in Omniglot 5-way 1-shot, as well as *mini*ImageNet 5-way 1-shot setting.

| Algorithm | Worst 10% | Worst 20% | Worst 30% | Worst 40% | Worst 50% | Overall |
|---|---|---|---|---|---|---|
| Omniglot 5-way 1-shot | | | | | | |
| MAML | $91.71_{\pm 0.73}$ | $94.16_{\pm 0.50}$ | $95.41_{\pm 0.39}$ | $96.22_{\pm 0.32}$ | $96.76_{\pm 0.27}$ | $98.38_{\pm 0.17}$ |
| Reptile | $82.78_{\pm 0.85}$ | $86.22_{\pm 0.64}$ | $88.33_{\pm 0.54}$ | $89.79_{\pm 0.48}$ | $90.93_{\pm 0.43}$ | $94.64_{\pm 0.32}$ |
| Protonet | $88.72_{\pm 0.99}$ | $92.24_{\pm 0.70}$ | $93.95_{\pm 0.54}$ | $95.06_{\pm 0.44}$ | $95.79_{\pm 0.38}$ | $97.82_{\pm 0.23}$ |
| Matching Networks | $79.70_{\pm 0.95}$ | $84.01_{\pm 0.78}$ | $86.78_{\pm 0.68}$ | $88.83_{\pm 0.62}$ | $90.41_{\pm 0.56}$ | $94.71_{\pm 0.39}$ |
| MAML + RGD | $\mathbf{92.14}_{\pm 0.84}$ | $\mathbf{94.54}_{\pm 0.53}$ | $\mathbf{95.72}_{\pm 0.40}$ | $\mathbf{96.46}_{\pm 0.33}$ | $\mathbf{96.90}_{\pm 0.27}$ | $\mathbf{98.45}_{\pm 0.17}$ |
| Omniglot 20-way 1-shot | | | | | | |
| MAML | $84.33_{\pm 0.40}$ | $85.86_{\pm 0.29}$ | $86.92_{\pm 0.26}$ | $87.73_{\pm 0.24}$ | $88.42_{\pm 0.22}$ | $91.28_{\pm 0.22}$ |
| Reptile | $83.13_{\pm 0.42}$ | $84.71_{\pm 0.31}$ | $85.77_{\pm 0.26}$ | $86.60_{\pm 0.24}$ | $87.30_{\pm 0.23}$ | $90.09_{\pm 0.22}$ |
| Protonet | $\mathbf{87.19}_{\pm 0.33}$ | $\mathbf{88.71}_{\pm 0.27}$ | $\mathbf{89.73}_{\pm 0.24}$ | $\mathbf{90.54}_{\pm 0.23}$ | $\mathbf{91.20}_{\pm 0.22}$ | $\mathbf{93.72}_{\pm 0.20}$ |
| Matching Networks | $62.82_{\pm 0.60}$ | $65.50_{\pm 0.48}$ | $67.25_{\pm 0.42}$ | $68.61_{\pm 0.39}$ | $69.75_{\pm 0.37}$ | $74.62_{\pm 0.38}$ |
| MAML + RGD | $86.61_{\pm 0.36}$ | $88.09_{\pm 0.28}$ | $89.09_{\pm 0.24}$ | $89.87_{\pm 0.23}$ | $90.50_{\pm 0.21}$ | $93.01_{\pm 0.20}$ |
| *mini*ImageNet 5-way 1-shot | | | | | | |
| MAML | $30.94_{\pm 0.70}$ | $34.52_{\pm 0.62}$ | $36.93_{\pm 0.57}$ | $38.94_{\pm 0.55}$ | $40.68_{\pm 0.53}$ | $48.86_{\pm 0.62}$ |
| Reptile | $25.37_{\pm 0.74}$ | $28.59_{\pm 0.59}$ | $30.71_{\pm 0.52}$ | $32.52_{\pm 0.50}$ | $34.11_{\pm 0.48}$ | $41.42_{\pm 0.56}$ |
| Protonet | $30.93_{\pm 0.76}$ | $34.62_{\pm 0.65}$ | $37.06_{\pm 0.58}$ | $38.94_{\pm 0.54}$ | $40.66_{\pm 0.52}$ | $48.56_{\pm 0.60}$ |
| Matching Networks | $27.19_{\pm 0.68}$ | $30.42_{\pm 0.57}$ | $32.64_{\pm 0.52}$ | $34.45_{\pm 0.50}$ | $36.10_{\pm 0.49}$ | $43.84_{\pm 0.58}$ |
| MAML + RGD | $\mathbf{33.33}_{\pm 0.90}$ | $\mathbf{36.67}_{\pm 0.65}$ | $\mathbf{39.12}_{\pm 0.59}$ | $\mathbf{41.20}_{\pm 0.56}$ | $\mathbf{42.96}_{\pm 0.55}$ | $\mathbf{51.21}_{\pm 0.63}$ |

Table 24: Results on DomainBed (Model selection: training-domain validation set): The bottom partition shows results of our method with RGD loss. In both cases, with (top) and without (bottom) fixed linear layer, the proposed approach outperforms existing methods, as well as SOTA.

| Algorithm | PACS | VLCS | OfficeHome | DomainNet | Avg. |
|---|---|---|---|---|---|
| ERM Gulrajani & Lopez-Paz (2021) | $85.5_{\pm 0.1}$ | $77.5_{\pm 0.4}$ | $66.5_{\pm 0.2}$ | $40.9_{\pm 0.1}$ | 67.6 |
| IRM Arjovsky et al. (2019) | $83.5_{\pm 0.8}$ | $78.5_{\pm 0.5}$ | $64.3_{\pm 2.2}$ | $33.9_{\pm 2.8}$ | 65.1 |
| GroupDRO Sagawa* et al. (2020) | $84.4_{\pm 0.8}$ | $76.7_{\pm 0.6}$ | $66.0_{\pm 0.7}$ | $33.3_{\pm 0.2}$ | 65.1 |
| Mixup Yan et al. (2020a) | $84.6_{\pm 0.6}$ | $77.4_{\pm 0.6}$ | $68.1_{\pm 0.3}$ | $39.2_{\pm 0.1}$ | 67.33 |
| MLDG Li et al. (2018a) | $84.9_{\pm 1.0}$ | $77.2_{\pm 0.4}$ | $66.8_{\pm 0.6}$ | $41.2_{\pm 0.1}$ | 67.53 |
| CORAL Sun & Saenko (2016) | $86.2_{\pm 0.3}$ | $78.8_{\pm 0.6}$ | $68.7_{\pm 0.3}$ | $41.5_{\pm 0.1}$ | 68.8 |
| MMD Li et al. (2018b) | $84.6_{\pm 0.5}$ | $77.5_{\pm 0.9}$ | $66.3_{\pm 0.1}$ | $23.4_{\pm 9.5}$ | 62.95 |
| DANN Ganin et al. (2016) | $83.6_{\pm 0.4}$ | $78.6_{\pm 0.4}$ | $65.9_{\pm 0.6}$ | $38.3_{\pm 0.1}$ | 66.6 |
| CDANN Li et al. (2018c) | $82.6_{\pm 0.9}$ | $77.5_{\pm 0.1}$ | $65.8_{\pm 1.3}$ | $38.3_{\pm 0.3}$ | 66.05 |
| MTL Blanchard et al. (2021) | $84.6_{\pm 0.5}$ | $77.2_{\pm 0.4}$ | $66.4_{\pm 0.5}$ | $40.6_{\pm 0.1}$ | 67.2 |
| SagNet Nam et al. (2021) | $86.3_{\pm 0.2}$ | $77.8_{\pm 0.5}$ | $68.1_{\pm 0.1}$ | $40.3_{\pm 0.1}$ | 68.13 |
| ARM Zhang et al. (2021) | $85.1_{\pm 0.4}$ | $77.6_{\pm 0.3}$ | $64.8_{\pm 0.3}$ | $35.5_{\pm 0.2}$ | 65.75 |
| VREx Krueger et al. (2021) | $84.9_{\pm 0.6}$ | $78.3_{\pm 0.2}$ | $66.4_{\pm 0.6}$ | $33.6_{\pm 2.9}$ | 65.8 |
| RSC Huang et al. (2020) | $85.2_{\pm 0.9}$ | $77.1_{\pm 0.5}$ | $65.5_{\pm 0.9}$ | $38.9_{\pm 0.5}$ | 66.68 |
| MIRO Cha et al. (2022) | $85.4_{\pm 0.4}$ | $79.0_{\pm 0.0}$ | $70.5_{\pm 0.4}$ | $44.3_{\pm 0.2}$ | 69.8 |
| ERM + FRR-L | | | | | |
| Default Addepalli et al. (2023) | $85.7_{\pm 0.1}$ | $76.6_{\pm 0.2}$ | $68.4_{\pm 0.2}$ | $44.2_{\pm 0.1}$ | 68.73 |
| RGD-RevKL (**Ours**) | $87.2_{\pm 0.3}$ | $78.6_{\pm 0.3}$ | $69.4_{\pm 0.2}$ | $45.8_{\pm 0.0}$ | 70.25 |
| RGD (**Ours**) | $\mathbf{87.6}_{\pm 0.3}$ | $78.6_{\pm 0.3}$ | $\mathbf{69.8}_{\pm 0.2}$ | $\mathbf{46.0}_{\pm 0.0}$ | **70.48** |
| ERM + FRR | | | | | |
| Default Addepalli et al. (2023) | $87.5_{\pm 0.1}$ | $77.6_{\pm 0.3}$ | $69.4_{\pm 0.1}$ | $45.1_{\pm 0.1}$ | 69.9 |
| RGD-RevKL (**Ours**) | $87.6_{\pm 0.3}$ | $78.1_{\pm 0.1}$ | $\mathbf{69.9}_{\pm 0.1}$ | $45.8_{\pm 0.0}$ | 70.35 |
| RGD (**Ours**) | $\mathbf{88.2}_{\pm 0.2}$ | $78.6_{\pm 0.3}$ | $69.8_{\pm 0.2}$ | $45.8_{\pm 0.0}$ | **70.6** |

Table 25: **Out-of-domain accuracies (%) on PACS.**

| Algorithm | A | C | P | S | Avg |
|---|---|---|---|---|---|
| CDANN | $84.6_{\pm 1.8}$ | $75.5_{\pm 0.9}$ | $96.8_{\pm 0.3}$ | $73.5_{\pm 0.6}$ | 82.6 |
| MASF | 82.9 | 80.5 | 95.0 | 72.3 | 82.7 |
| DMG | 82.6 | 78.1 | 94.5 | 78.3 | 83.4 |
| IRM | $84.8_{\pm 1.3}$ | $76.4_{\pm 1.1}$ | $96.7_{\pm 0.6}$ | $76.1_{\pm 1.0}$ | 83.5 |
| MetaReg | 87.2 | 79.2 | 97.6 | 70.3 | 83.6 |
| DANN | $86.4_{\pm 0.8}$ | $77.4_{\pm 0.8}$ | $97.3_{\pm 0.4}$ | $73.5_{\pm 2.3}$ | 83.7 |
| GroupDRO | $83.5_{\pm 0.9}$ | $79.1_{\pm 0.6}$ | $96.7_{\pm 0.3}$ | $78.3_{\pm 2.0}$ | 84.4 |
| MTL | $87.5_{\pm 0.8}$ | $77.1_{\pm 0.5}$ | $96.4_{\pm 0.8}$ | $77.3_{\pm 1.8}$ | 84.6 |
| I-Mixup | $86.1_{\pm 0.5}$ | $78.9_{\pm 0.8}$ | $97.6_{\pm 0.1}$ | $75.8_{\pm 1.8}$ | 84.6 |
| MMD | $86.1_{\pm 1.4}$ | $79.4_{\pm 0.9}$ | $96.6_{\pm 0.2}$ | $76.5_{\pm 0.5}$ | 84.7 |
| VREx | $86.0_{\pm 1.6}$ | $79.1_{\pm 0.6}$ | $96.9_{\pm 0.5}$ | $77.7_{\pm 1.7}$ | 84.9 |
| MLDG | $85.5_{\pm 1.4}$ | $80.1_{\pm 1.7}$ | $97.4_{\pm 0.3}$ | $76.6_{\pm 1.1}$ | 84.9 |
| ARM | $86.8_{\pm 0.6}$ | $76.8_{\pm 0.5}$ | $97.4_{\pm 0.3}$ | $79.3_{\pm 1.2}$ | 85.1 |
| RSC | $85.4_{\pm 0.8}$ | $79.7_{\pm 1.8}$ | $97.6_{\pm 0.3}$ | $78.2_{\pm 1.2}$ | 85.2 |
| Mixstyle | $86.8_{\pm 0.5}$ | $79.0_{\pm 1.4}$ | $96.6_{\pm 0.1}$ | $78.5_{\pm 2.3}$ | 85.2 |
| ER | 87.5 | 79.3 | 98.3 | 76.3 | 85.3 |
| pAdaIN | 85.8 | 81.1 | 97.2 | 77.4 | 85.4 |
| ERM | $84.7_{\pm 0.4}$ | $80.8_{\pm 0.6}$ | $97.2_{\pm 0.3}$ | $79.3_{\pm 1.0}$ | 85.5 |
| EISNet | 86.6 | 81.5 | 97.1 | 78.1 | 85.8 |
| CORAL | $88.3_{\pm 0.2}$ | $80.0_{\pm 0.5}$ | $97.5_{\pm 0.3}$ | $78.8_{\pm 1.3}$ | 86.2 |
| SagNet | $87.4_{\pm 1.0}$ | $80.7_{\pm 0.6}$ | $97.1_{\pm 0.1}$ | $80.0_{\pm 0.4}$ | 86.3 |
| DSON | 87.0 | 80.6 | 96.0 | 82.9 | 86.6 |
| ERM + FRR-L | | | | | |
| Default | $83.2_{\pm 0.3}$ | $79.8_{\pm 0.4}$ | $95.9_{\pm 0.3}$ | $83.5_{\pm 0.4}$ | 85.7 |
| RGD-RevKL | $88.7_{\pm 0.5}$ | $83.0_{\pm 0.5}$ | $97.8_{\pm 0.1}$ | $79.4_{\pm 1.0}$ | 87.2 |
| RGD | $88.4_{\pm 0.3}$ | $83.3_{\pm 0.8}$ | $97.5_{\pm 0.3}$ | $81.1_{\pm 0.5}$ | **87.6** |
| ERM + FRR | | | | | |
| Default | $86.8_{\pm 0.3}$ | $82.2_{\pm 0.4}$ | $96.4_{\pm 0.1}$ | $84.5_{\pm 0.2}$ | 87.5 |
| RGD-RevKL | $87.7_{\pm 0.8}$ | $84.0_{\pm 0.6}$ | $97.6_{\pm 0.1}$ | $81.2_{\pm 0.5}$ | 87.6 |
| RGD | $88.8_{\pm 0.3}$ | $84.0_{\pm 0.8}$ | $97.7_{\pm 0.1}$ | $82.4_{\pm 0.6}$ | **88.2** |

## D.9 Convergence of RGD and additional costs

**Convergence in extreme class imbalance setting:** In the extreme class imbalance setting, we note that uniform sampling for mini-batch generation + RGD re-weighting (as done in Algorithm 1) would be slower to converge than using importance sampling for mini-batch generation. This is because the former tends to have higher variance. But this is easily fixable in Algorithm 1. We simply update the mini-batch generation step with importance sampling; that is, we select point '$i$' with probability proportional to its weight $\exp(\ell_i)$ (instead of uniform sampling that is currently done). The main reason for not considering this in this work is our desire to illustrate the generality of our approach and its applicability to a wide variety of learning tasks, without focusing too much on the class imbalance task. We believe this generality and simplicity is what makes our method quite attractive to the practitioner as showcased in some of experiments including Natural Language Processing, Image Classification, Tabular Classification, Distribution Shifts, and Meta-learning. Furthermore, Figure 6 illustrates the convergence plots of miniGPT pre-training and ViT-S on Imagenet-1K. Overall, we note similar stable training convergence on both, while RGD is able to focus more heavily on harder samples and reach a better minima.

Table 26: **Out-of-domain accuracies (%) on** VLCS.

| Algorithm | C | L | S | V | Avg |
|---|---|---|---|---|---|
| GroupDRO | 97.3 ± 0.3 | 63.4 ± 0.9 | 69.5 ± 0.8 | 76.7 ± 0.7 | 76.7 |
| RSC | 97.9 ± 0.1 | 62.5 ± 0.7 | 72.3 ± 1.2 | 75.6 ± 0.8 | 77.1 |
| MLDG | 97.4 ± 0.2 | 65.2 ± 0.7 | 71.0 ± 1.4 | 75.3 ± 1.0 | 77.2 |
| MTL | 97.8 ± 0.4 | 64.3 ± 0.3 | 71.5 ± 0.7 | 75.3 ± 1.7 | 77.2 |
| I-Mixup | 98.3 ± 0.6 | 64.8 ± 1.0 | 72.1 ± 0.5 | 74.3 ± 0.8 | 77.4 |
| ERM | 97.7 ± 0.4 | 64.3 ± 0.9 | 73.4 ± 0.5 | 74.6 ± 1.3 | 77.5 |
| MMD | 97.7 ± 0.1 | 64.0 ± 1.1 | 72.8 ± 0.2 | 75.3 ± 3.3 | 77.5 |
| CDANN | 97.1 ± 0.3 | 65.1 ± 1.2 | 70.7 ± 0.8 | 77.1 ± 1.5 | 77.5 |
| ARM | 98.7 ± 0.2 | 63.6 ± 0.7 | 71.3 ± 1.2 | 76.7 ± 0.6 | 77.6 |
| SagNet | 97.9 ± 0.4 | 64.5 ± 0.5 | 71.4 ± 1.3 | 77.5 ± 0.5 | 77.8 |
| Mixstyle | 98.6 ± 0.3 | 64.5 ± 1.1 | 72.6 ± 0.5 | 75.7 ± 1.7 | 77.9 |
| VREx | 98.4 ± 0.3 | 64.4 ± 1.4 | 74.1 ± 0.4 | 76.2 ± 1.3 | 78.3 |
| IRM | 98.6 ± 0.1 | 64.9 ± 0.9 | 73.4 ± 0.6 | 77.3 ± 0.9 | 78.6 |
| DANN | 99.0 ± 0.3 | 65.1 ± 1.4 | 73.1 ± 0.3 | 77.2 ± 0.6 | 78.6 |
| CORAL | 98.3 ± 0.1 | 66.1 ± 1.2 | 73.4 ± 0.3 | 77.5 ± 1.2 | 78.8 |
| ERM + FRR-L | | | | | |
| Default | 97.1 ± 0.2 | 63.3 ± 0.3 | 72.0 ± 0.3 | 74.3 ± 0.3 | 76.6 |
| RGD-RevKL | 98.8 ± 0.1 | 64.8 ± 0.2 | 73.9 ± 0.2 | 77.0 ± 1.1 | 78.6 |
| RGD | 98.9 ± 0 | 64.9 ± 0.4 | 73.2 ± 0.4 | 77.5 ± 0.6 | 78.6 |
| ERM + FRR | | | | | |
| Default | 96.7 ± 0.6 | 65.2 ± 0.8 | 73.4 ± 0.1 | 75.6 ± 0.4 | 77.6 |
| RGD-RevKL | 98.3 ± 0.1 | 64.5 ± 0.2 | 72.3 ± 0.1 | 77.2 ± 0.3 | 78.1 |
| RGD | 97.1 ± 0.5 | 65.4 ± 0.8 | 74.3 ± 0.1 | 77.5 ± 0.3 | 78.6 |

**Additional costs of RGD:** Furthermore, we note that RGD poses **NO** additional cost over standard approaches. The approach is a simple modification of the loss with a closed-form function with $\mathcal{O}(1)$ complexity, and without any changes in architecture, training regime, etc.

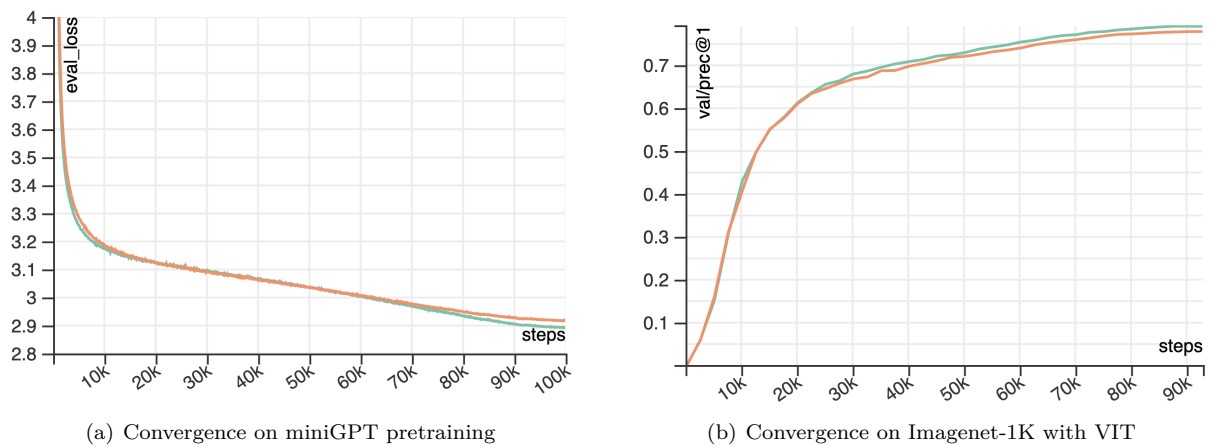

(a) Convergence on miniGPT pretraining

(b) Convergence on Imagenet-1K with VIT

Figure 6: Convergence plots of RGD and Default training regime on real-world datasets. Here the orange line denotes the default training regime, and the green line denotes RGD.

Table 27: **Out-of-domain accuracies (%) on OfficeHome.**

| Algorithm | A | C | P | R | Avg |
|---|---|---|---|---|---|
| Mixstyle | $51.1 \pm 0.3$ | $53.2 \pm 0.4$ | $68.2 \pm 0.7$ | $69.2 \pm 0.6$ | 60.4 |
| IRM | $58.9 \pm 2.3$ | $52.2 \pm 1.6$ | $72.1 \pm 2.9$ | $74.0 \pm 2.5$ | 64.3 |
| ARM | $58.9 \pm 0.8$ | $51.0 \pm 0.5$ | $74.1 \pm 0.1$ | $75.2 \pm 0.3$ | 64.8 |
| RSC | $60.7 \pm 1.4$ | $51.4 \pm 0.3$ | $74.8 \pm 1.1$ | $75.1 \pm 1.3$ | 65.5 |
| CDANN | $61.5 \pm 1.4$ | $50.4 \pm 2.4$ | $74.4 \pm 0.9$ | $76.6 \pm 0.8$ | 65.7 |
| DANN | $59.9 \pm 1.3$ | $53.0 \pm 0.3$ | $73.6 \pm 0.7$ | $76.9 \pm 0.5$ | 65.9 |
| GroupDRO | $60.4 \pm 0.7$ | $52.7 \pm 1.0$ | $75.0 \pm 0.7$ | $76.0 \pm 0.7$ | 66.0 |
| MMD | $60.4 \pm 0.2$ | $53.3 \pm 0.3$ | $74.3 \pm 0.1$ | $77.4 \pm 0.6$ | 66.4 |
| MTL | $61.5 \pm 0.7$ | $52.4 \pm 0.6$ | $74.9 \pm 0.4$ | $76.8 \pm 0.4$ | 66.4 |
| VREx | $60.7 \pm 0.9$ | $53.0 \pm 0.9$ | $75.3 \pm 0.1$ | $76.6 \pm 0.5$ | 66.4 |
| ERM | $61.3 \pm 0.7$ | $52.4 \pm 0.3$ | $75.8 \pm 0.1$ | $76.6 \pm 0.3$ | 66.5 |
| MLDG | $61.5 \pm 0.9$ | $53.2 \pm 0.6$ | $75.0 \pm 1.2$ | $77.5 \pm 0.4$ | 66.8 |
| I-Mixup | $62.4 \pm 0.8$ | $54.8 \pm 0.6$ | $76.9 \pm 0.3$ | $78.3 \pm 0.2$ | 68.1 |
| SagNet | $63.4 \pm 0.2$ | $54.8 \pm 0.4$ | $75.8 \pm 0.4$ | $78.3 \pm 0.3$ | 68.1 |
| CORAL | $65.3 \pm 0.4$ | $54.4 \pm 0.5$ | $76.5 \pm 0.1$ | $78.4 \pm 0.5$ | 68.7 |
| ERM + FRR-L | | | | | |
| Default | $64.4 \pm 0.1$ | $55.6 \pm 0.5$ | $76.5 \pm 0.2$ | $77.5 \pm 0.2$ | 68.4 |
| RGD-RevKL | $64.2 \pm 0.3$ | $55.9 \pm 0.5$ | $77.6 \pm 0.2$ | $79.9 \pm 0.3$ | 69.4 |
| RGD | $64.5 \pm 0.3$ | $56.9 \pm 0.5$ | $77.8 \pm 0.3$ | $80.0 \pm 0.4$ | **69.8** |
| ERM + FRR | | | | | |
| Default | $64.5 \pm 0.2$ | $58.4 \pm 0.1$ | $76.6 \pm 0.3$ | $78.3 \pm 0.1$ | 69.4 |
| RGD-RevKL | $65.6 \pm 0.3$ | $57.1 \pm 0.3$ | $76.8 \pm 0.3$ | $80.2 \pm 0.2$ | **69.9** |
| RGD | $65.6 \pm 0.5$ | $56.9 \pm 0.3$ | $76.9 \pm 0.1$ | $79.7 \pm 0.3$ | 69.8 |

Table 28: **Out-of-domain accuracies (%) on** `DomainNet`.

| Algorithm | clip | info | paint | quick | real | sketch | Avg |
|---|---|---|---|---|---|---|---|
| MMD | $32.1_{\pm 13.3}$ | $11.0_{\pm 4.6}$ | $26.8_{\pm 11.3}$ | $8.7_{\pm 2.1}$ | $32.7_{\pm 13.8}$ | $28.9_{\pm 11.9}$ | 23.4 |
| GroupDRO | $47.2_{\pm 0.5}$ | $17.5_{\pm 0.4}$ | $33.8_{\pm 0.5}$ | $9.3_{\pm 0.3}$ | $51.6_{\pm 0.4}$ | $40.1_{\pm 0.6}$ | 33.3 |
| VREx | $47.3_{\pm 3.5}$ | $16.0_{\pm 1.5}$ | $35.8_{\pm 4.6}$ | $10.9_{\pm 0.3}$ | $49.6_{\pm 4.9}$ | $42.0_{\pm 3.0}$ | 33.6 |
| IRM | $48.5_{\pm 2.8}$ | $15.0_{\pm 1.5}$ | $38.3_{\pm 4.3}$ | $10.9_{\pm 0.5}$ | $48.2_{\pm 5.2}$ | $42.3_{\pm 3.1}$ | 33.9 |
| Mixstyle | $51.9_{\pm 0.4}$ | $13.3_{\pm 0.2}$ | $37.0_{\pm 0.5}$ | $12.3_{\pm 0.1}$ | $46.1_{\pm 0.3}$ | $43.4_{\pm 0.4}$ | 34.0 |
| ARM | $49.7_{\pm 0.3}$ | $16.3_{\pm 0.5}$ | $40.9_{\pm 1.1}$ | $9.4_{\pm 0.1}$ | $53.4_{\pm 0.4}$ | $43.5_{\pm 0.4}$ | 35.5 |
| CDANN | $54.6_{\pm 0.4}$ | $17.3_{\pm 0.1}$ | $43.7_{\pm 0.9}$ | $12.1_{\pm 0.7}$ | $56.2_{\pm 0.4}$ | $45.9_{\pm 0.5}$ | 38.3 |
| DANN | $53.1_{\pm 0.2}$ | $18.3_{\pm 0.1}$ | $44.2_{\pm 0.7}$ | $11.8_{\pm 0.1}$ | $55.5_{\pm 0.4}$ | $46.8_{\pm 0.6}$ | 38.3 |
| RSC | $55.0_{\pm 1.2}$ | $18.3_{\pm 0.5}$ | $44.4_{\pm 0.6}$ | $12.2_{\pm 0.2}$ | $55.7_{\pm 0.7}$ | $47.8_{\pm 0.9}$ | 38.9 |
| I-Mixup | $55.7_{\pm 0.3}$ | $18.5_{\pm 0.5}$ | $44.3_{\pm 0.5}$ | $12.5_{\pm 0.4}$ | $55.8_{\pm 0.3}$ | $48.2_{\pm 0.5}$ | 39.2 |
| SagNet | $57.7_{\pm 0.3}$ | $19.0_{\pm 0.2}$ | $45.3_{\pm 0.3}$ | $12.7_{\pm 0.5}$ | $58.1_{\pm 0.5}$ | $48.8_{\pm 0.2}$ | 40.3 |
| MTL | $57.9_{\pm 0.5}$ | $18.5_{\pm 0.4}$ | $46.0_{\pm 0.1}$ | $12.5_{\pm 0.1}$ | $59.5_{\pm 0.3}$ | $49.2_{\pm 0.1}$ | 40.6 |
| ERM | $58.1_{\pm 0.3}$ | $18.8_{\pm 0.3}$ | $46.7_{\pm 0.3}$ | $12.2_{\pm 0.4}$ | $59.6_{\pm 0.1}$ | $49.8_{\pm 0.4}$ | 40.9 |
| MLDG | $59.1_{\pm 0.2}$ | $19.1_{\pm 0.3}$ | $45.8_{\pm 0.7}$ | $13.4_{\pm 0.3}$ | $59.6_{\pm 0.2}$ | $50.2_{\pm 0.4}$ | 41.2 |
| CORAL | $59.2_{\pm 0.1}$ | $19.7_{\pm 0.2}$ | $46.6_{\pm 0.3}$ | $13.4_{\pm 0.4}$ | $59.8_{\pm 0.2}$ | $50.1_{\pm 0.6}$ | 41.5 |
| MetaReg | 59.8 | 25.6 | 50.2 | 11.5 | 64.6 | 50.1 | 43.6 |
| DMG | 65.2 | 22.2 | 50.0 | 15.7 | 59.6 | 49.0 | 43.6 |
| **ERM + FRR-L** | | | | | | | |
| Default | $63.6_{\pm 0.1}$ | $20.5_{\pm 0.0}$ | $50.7_{\pm 0.0}$ | $14.6_{\pm 0.1}$ | $63.8_{\pm 0.1}$ | $53.4_{\pm 0.0}$ | 44.2 |
| RGD-RevKL | $65.7_{\pm 0.1}$ | $21.9_{\pm 0.0}$ | $52.0_{\pm 0.1}$ | $15.1_{\pm 0.1}$ | $65.2_{\pm 0.1}$ | $54.9_{\pm 0.1}$ | 45.8 |
| RGD | $65.8_{\pm 0.1}$ | $22.1_{\pm 0.0}$ | $52.3_{\pm 0.1}$ | $15.1_{\pm 0.1}$ | $65.7_{\pm 0.0}$ | $54.8_{\pm 0.1}$ | **46.0** |
| **ERM + FRR** | | | | | | | |
| Default | $64.3_{\pm 0.1}$ | $21.2_{\pm 0.3}$ | $51.1_{\pm 0.2}$ | $14.9_{\pm 0.6}$ | $64.7_{\pm 0.1}$ | $54.1_{\pm 0.2}$ | 45.1 |
| RGD-RevKL | $65.6_{\pm 0.0}$ | $21.9_{\pm 0.0}$ | $52.0_{\pm 0.1}$ | $15.0_{\pm 0.1}$ | $65.5_{\pm 0.0}$ | $54.8_{\pm 0.1}$ | 45.8 |
| RGD | $65.6_{\pm 0.0}$ | $21.5_{\pm 0.0}$ | $52.1_{\pm 0.0}$ | $15.0_{\pm 0.0}$ | $65.7_{\pm 0.0}$ | $55.1_{\pm 0.0}$ | 45.8 |

Table 29: Results on DomainBed (Model selection: training-domain validation set) on the model weight averaging models such as SWAD Cha et al. (2021): The bottom partition shows results of our method with RGD loss.

| Algorithm | PACS | VLCS | OfficeHome | DomainNet | Avg. |
|---|---|---|---|---|---|
| SWAD | | | | | |
| Default Cha et al. (2021) | $86.5_{\pm 1.0}$ | $\mathbf{76.0}_{\pm 0.7}$ | $66.3_{\pm 0.2}$ | $43.8_{\pm 0.1}$ | 68.15 |
| RGD (**Ours**) | $\mathbf{87.6}_{\pm 0.2}$ | $75.4_{\pm 1.1}$ | $\mathbf{67.5}_{\pm 0.3}$ | $\mathbf{44.0}_{\pm 0.1}$ | **68.63** |

