# OpenReview forum: "Stochastic Re-weighted Gradient Descent via Distributionally Robust Optimization"
_TMLR — Accepted by TMLR_

### Review · Reviewer_br2F · 2024-07-21

**Summary Of Contributions:**

This paper investigates the use of distributionally robust optimization for optimizing the parameters in many recent ML settings (supervised learning for language/vision, tabular classification, domain generalization, and meta-learning). The basic formulation for DRO here is given in S3.2 and in Prop 3.1 where they add a KL-based uncertainty around the data, and Section 1.1 describes the main experimental setups. The experimental results throughout section 4 consistently show the advantages of the approach.

**Audience:**

Yes

**Claims And Evidence:**

Yes

**Requested Changes:**

The paper is good, I do not have any requested changes other than responding to my questions on the weaknesses.

**Strengths And Weaknesses:**

**Strengths**
+ The use of DRO in this setting is clearly motivated as the training data is clearly noise in some way, and the experiments clearly document and show the performance gains of Alg 1 over the standard updates.

**Weaknesses**
+ Could some of the performance gains come from differences in how the hyper-parameters of the optimizer are searched over? In many of the settings, it's not clear, for example, if the baselines and RGD versions 1) use the same optimizer, and 2) search over the hyper-parameters in the same way for each optimizer. While the paper mentions Adam could be used in place of the SGD described in the proposition, I think it is important to be clear about what optimizers the baselines and RGD methods are based on
+ There is not much of a discussion on the limitations here. If the method works as well as the paper makes it seem, then it seems like there would be little reason for almost every paper not to use DRO and re-weighted variants for parameter optimization instead of just running Adam --- is it true, or are there other limitations and factors to consider?

---

> ### Author Response · Authors · 2024-08-16
>
> We thank the reviewer for their positive feedback and insightful questions. We address the raised concerns regarding hyperparameter tuning and limitations below:
>
> > Could some of the performance gains come from differences in how the hyper-parameters of the optimizer are searched over? In many of the settings, it's not clear, for example, if the baselines and RGD versions 1) use the same optimizer, and 2) search over the hyper-parameters in the same way for each optimizer. While the paper mentions Adam could be used in place of the SGD described in the proposition, I think it is important to be clear about what optimizers the baselines and RGD methods are based on
>
> - To ensure fair comparisons, we used the same optimizer for both baseline and RGD. In all our experiments, we simply integrated RGD into the baseline codebases, maintaining the same optimizer (primarily Adam for most experiments, SGD for CIFAR experiments) for both baseline and RGD versions.  RGD introduces only one additional hyperparameter, the clipping factor ($\tau$).
>
> For all experiments:
> * **Baseline:** We tuned the learning rate (lr).
> * **RGD:** We tuned both $\tau$ and lr.
>
> Appendix D.1 provides details of the hyperparameter search spaces used.  We emphasize that RGD is a meta-optimization technique that *operates on top* of the existing optimizer.  It re-scales the per-sample loss before the optimizer's gradient update step, but it does not require changing the underlying optimizer.
>
> > There is not much of a discussion on the limitations here. If the method works as well as the paper makes it seem, then it seems like there would be little reason for almost every paper not to use DRO and re-weighted variants for parameter optimization instead of just running Adam --- is it true, or are there other limitations and factors to consider?
> - While RGD shows promising results, it has the following limitations that warrant further investigation. We note that we briefly mentioned these points in the conclusion and future work section of our paper:
>
> * **Outlier Handling:** Our current approach uses weight clipping to mitigate the impact of outliers. While empirically effective, a more principled approach to outlier robustness in DRO is worth exploring.
> * **Robustness to Noise:** While RGD can handle benign noise, it can fail in the presence of adversarial/systematic noise. Future work should investigate the performance and robustness of RGD in such settings, potentially exploring strategies that perform outlier removal before applying RGD.
>
>
> The reviewer raises the valid point that if RGD were universally superior, it would be widely adopted.  We believe that:
>
> - **Novelty and Empirical Validation:** Our work is the first to systematically demonstrate the effectiveness of KL-DRO-based re-weighting across diverse tasks and large-scale settings, achieving state-of-the-art results in several domains. This empirical validation is a significant contribution.
> - **Opening New Research Directions:** We believe our work also opens up potential directions for future research in outlier robust DRO.
>
> We are excited about the potential of RGD and believe it represents a promising step towards more robust and efficient training of deep neural networks.

---

> > ### Comment · Reviewer_br2F · 2024-08-18
> >
> > Thank you for the response! I have read through the other reviews and responses, and do not have any outstanding concerns over the hyperparameters or limitations

---

### Review · Reviewer_sqSP · 2024-08-06

**Summary Of Contributions:**

This paper introduces a novel re-weighted gradient descent technique designed to enhance the training of deep neural networks. The core concept is based on distributionally robust optimization, which helps the proposed algorithm efficiently manage optimization efficiency. Additionally, it shows promise in mitigating the impact of outliers, thereby stabilizing the stochastic gradient descent SGD algorithm.

**Audience:**

Yes

**Claims And Evidence:**

Yes

**Requested Changes:**

Report confidence interval of metrics in Tables 2-6.

**Strengths And Weaknesses:**

Strengths:

+ The paper effectively avoids the need to solve costly min-max optimization problems by deriving an equivalent optimization objective similar to titled empirical risk minimization. This objective can be efficiently addressed using stochastic gradient descent (SGD).


+ The proposed re-weighting technique is thoroughly evaluated across various tasks, including supervised learning, classification, and meta-learning, demonstrating the broad generalizability of the method.


+ The method is supported by strong theoretical results, although the proof is currently limited to the convex case.
Weakness:

Weakness:

- The reported metrics in Section 4 lack confidence intervals, making it difficult to assess the variability in model performance.

---

> ### Author Response · Authors · 2024-08-16
>
> We thank the reviewer for their insightful comments and appreciation of our work. We address the request for confidence intervals below:
>
> - **Table 2:** Here is the updated Table 2 with the confidence intervals based on 3 random seeds.
>
> | **Algorithm** | **ERM**          | **TERM**         | **ABSGD**        | **RGD**          |
> |---------------|------------------|------------------|------------------|------------------|
> | Accuracy      | 78.44 $\pm$ 0.35 | 78.03 $\pm$ 0.27 | 77.12 $\pm$ 0.82 | 79.11 $\pm$ 0.12 |
>
>
> - **Table 3:** Here is the updated Table 3 with confidence intervals based on 5 different seeds
>
> | Dataset                  | CIFAR-10 100 | CIFAR-10 10 | CIFAR-10 Avg. | CIFAR-100 100 | CIFAR-100 10 | CIFAR-100 Avg. |
> |--------------------------|--------------|-------------|---------------|---------------|--------------|----------------|
> | **Cross Entropy (CE)**    |              |             |               |               |              |                |
> | Default                   | 71.75 $\pm$ 0.75 | 87.64 $\pm$ 0.45| 79.70         | 38.35 $\pm$ 0.41  | 56.91 $\pm$ 0.41 | 47.63          |
> | TERM  | 72.20 $\pm$ 0.18 | 87.52 $\pm$ 0.07| 79.86         | 39.75 $\pm$ 0.04  | 57.91 $\pm$ 0.39 | 48.83          |
> | ABSGD | 72.43 $\pm$ 0.31 | **87.93 $\pm$ 0.25**| 80.18  | 39.77 $\pm$ 0.34  | 57.44 $\pm$ 0.25 | 48.61          |
> | **Ours**                  | **73.71 $\pm$ 0.04** | 87.78 $\pm$ 0.15| **80.75**   | **41.47 $\pm$ 0.30** | **58.49 $\pm$ 0.04** | **49.98**  |
>
>
> - **GLUE Benchmark** For the GLUE benchmark, we note that RGD outperforms the baseline by +1.94% with a standard deviation of 0.42% (computed over 5 random seeds), and is indeed statistically significant with a p-value that is less than 0.05. We will update the draft with this information.
>
> - **BERT Pretraining** For BERT pretraining, due to compute and time constraints, we only managed to run RGD for multiple seeds, but not the baseline. Here are the numbers for RGD along with standard deviation: Next Sequence Prediction (NSP) accuracy: 98.92 +/- 0.09, MLM accuracy: 71.35 +/- 0.11.
>
> Furthermore, we would like to highlight that the numbers provided in Table 5,6 are already averaged over 5 random seeds. Unfortunately, due to compute constraints, we are unable to run each experiment for multiple seeds **again** as they usually take weeks of training.
>
> We believe that:
> - The breadth of our evaluation across diverse tasks and domains, combined with strong theoretical foundations, provides compelling evidence for the effectiveness of RGD.
> - The inclusion of confidence intervals where feasible demonstrates our commitment to rigorous evaluation.

---

### Review · Reviewer_KKmY · 2024-08-12

**Summary Of Contributions:**

- New reweighting scheme inspired by Distributionally Robust Optimization (DRO)
- This method has wide applicability and is simple to adapt and implement.
- It is evaluated on a wide range of tasks and datasets, and is shown to improve performance over standard training methods.

**Audience:**

Yes

**Claims And Evidence:**

Yes

**Requested Changes:**

## Minor changes
- Update the bibliography to make sure all published works are referenced correctly (eg for TERM the arXiv paper is referenced while it is published at ICML 2021, ABSGD is published at TMLR etc...)
- Discuss the points in Related work

## Major changes
- Clarify the paragraph "Generalization" in \S 3.1
- Address the question about robustness to outlier (eg by experimenting with an artificially corrupted dataset, even on a synthetic task) or tone down this claim
- Address the reproducibility issue

## Enhancements
- Address the comparison with other performance enhancing techniques, at least SAM and at least with a discussion

**Strengths And Weaknesses:**

## Strengths
- Both theoretically grounded and simple method, that has wide applicability.
- The weight clipping to reduce the influence of outliers (and maybe control the variance of the exponential?) is a great idea.
- Broad and thorough evaluation across varied tasks, including actual deep learning setups: this is rare for a DRO method to scale up to the full training of a transformer model!

## Weaknesses

### Comparison to other optimization methods that improve generalization
If I understand correctly, this method is inspired by Robust Optimization but its main purported benefit is to improve general performance of general deep learning models. Though I agree that the litterature on this subject is immense, I believe a comparison to at least a few mainstream methods is warranted; for example, Sharpness-aware minimization: it is an optimization method (with links to robust optimization as well) that has been shown to improve generalization of deep learning models.

### Robustness to outlier

This work claims multiple times that the weight clipping makes the method more robust to outlier (eg, in contributions "We further enhance robustness with weight clipping to mitigate the influence of outlier"). However, this precise point is never actually demonstrated in the experiments.

### Paragraph "Generalization" in \S 3.1
I don't understand how one obtains the conclusion of this paragraph "the above result shows that the empirical DRO risk is an upper-bound on the population risk": is the constant $c_1$ in (2) the same as the one in the last equation? is $c_3$ greater than $c_2$?
Moreover, why would it imply that minimizing the empirical DRO risk would acchieve a better bias-variance tradeoff?

### Related work

- TERM seems like an extremely similar method, maybe a slightly more detailed comparison would be useful (if I understand correclty the main differences are the weighg clipping and the removal of the logarithm)

- No mention of Superquantile approaches in the related work, see eg
  Mehta, Ronak R. et al. “Stochastic Optimization for Spectral Risk Measures.” International Conference on Artificial Intelligence and Statistics (2022).
  Levy, Daniel, et al. "Large-scale methods for distributionally robust optimization." Advances in Neural Information Processing Systems 33 (2020): 8847-8860.
  (Though these methods do not scale to the true DL regime, I believe they should be mentionned)

### Reproducibility and experimental details
Many experimental details are missing, which I believe are important for transparency and reproducibility. For example:
- Which optimizer is used with RGD? It is plain SGD as in algorithm 1 or are the standard optimizers for each task (eg Adam) used?
- What about learning rate schedules? Weight decay? Batch-sizes? Stopping criterion? Everything is the same as in the original code bases?
- You mention that "Our experiments are based on public datasets and open-srouce code repositories", maybe citing the code / repo used would be a first step?

---

> ### Author Response · Authors · 2024-08-16
>
> We sincerely appreciate the reviewer's detailed feedback and insightful suggestions. We address each point raised below:
>
> > Comparison to Other Optimization Methods such as SAM
>
> -  Given that DRO and data reweighting are huge disciplines, we focused on these two areas in our paper. But we concur with the reviewer that comparisons with other generalization-enhancing optimization methods are important. Below we compare with Sharpness-Aware Minimization (SAM) both from theoretical and empirical perspectives. From a theoretical perspective, SAM performs robust optimization in the weight space (that is SAM tries to learn a model that is robust to perturbations of weights). In contrast, RGD performs robust optimization in the distribution space. So, RGD and SAM are orthogonal to each other and can potentially be merged together to boost the performance. Below, we compare performance of various approaches for CIFAR-10 optimization. It can be seen that RGD marginally boosts the performance of SAM. Although, more thorough experiments are needed to understand the utility of RGD on top of SAM.
>
> | **Algorithm** | **SGD**          | **SAM**           | **SAM + RGD**     |
> |---------------|------------------|-------------------|-------------------|
> | Accuracy      | 96.80 $\pm$ 0.04 | 97.31 $\pm$  0.05 | 97.41 $\pm$  0.09 |
>
> From a computational perspective, SAM is 2x more costly than the base optimizer (as it requires two forward passes in each step). Whereas RGD doesn’t add any computational overhead.
>
> > Clarify the paragraph "Generalization" in \S 3.1
> - Note that c_2/n in Equation 2 and c_3/n in the last equation of Section 3.1 are lower order terms. Modulo these lower order terms, it can be seen that the RHS in Equation 2 is nothing but the empirical DRO objective. So, empirical DRO is an upper bound for the population risk. We will clarify this in our paper. We would also like to refer the reviewer to Namkoong & Duchi, 2017 for a detailed discussion on this topic.
>
> > why would it imply that minimizing the empirical DRO risk would acchieve a better bias-variance tradeoff?
> - Based on the last equation in Section 3.1, it can be seen that empirical DRO is equal to the empirical risk  plus a variance term (modulo the lower order term). This variance term acts as a regularizer during the optimization of empirical DRO and leads to models with smaller variance.
>
> > Update the bibliography to make sure all published works are referenced correctly (eg for TERM the arXiv paper is referenced while it is published at ICML 2021, ABSGD is published at TMLR etc...)
>
> - As suggested, we will update the draft with the ICML, and TMLR for TERM and ABSGD respectively.
>
> > Address the question about robustness to outlier.
>
> - We thank the reviewer for the suggestion. To demonstrate that our technique can effectively handle benign outliers, we perform the following experiment. We randomly flip the labels in CIFAR-10, CIFAR-100 datasets (we vary the proportion of flips from 0% to 40%) and compare the performance of RGD with TERM. The results from this experiment are reported below. Note that existing DRO techniques which do not account for outliers perform poorly in this experiment, as they upweight the corrupted/mislabeled  points. Whereas, RGD which clips the weights of outliers, achieves the best performance.
>
> | Dataset                | CIFAR-10 0% | CIFAR-10 20% | CIFAR-10 40% | CIFAR-10 Avg. | CIFAR-100 0% | CIFAR-100 20% | CIFAR-100 40% | CIFAR-100 Avg. |
> |------------------------|-------------|--------------|--------------|---------------|--------------|---------------|---------------|----------------|
> | **Focal Loss**         | 93.03 ± 0.16 | 86.45 ± 0.19 | 80.45 ± 0.97 | 86.64         | 70.02 ± 0.53 | 61.87 ± 0.30  | 54.13 ± 0.40  | 62.01          |
> | **D2L**                | 92.02 ± 0.14 | 87.66 ± 0.40 | 83.89 ± 0.46 | 87.86         | 68.11 ± 0.26 | 63.48 ± 0.53  | 51.83 ± 0.33  | 61.14          |
> | **Cross Entropy (CE)** |             |              |              |               |              |               |               |                |
> | Default                | 92.89 ± 0.32 | 76.83 ± 2.30 | 70.77 ± 2.31 | 80.16         | 70.50 ± 0.12 | 50.86 ± 0.27  | 43.01 ± 1.16  | 54.79          |
> | TERM                   | 92.90 ± 0.09 | 58.70 ± 39.76| 73.17 ± 31.59| 74.92         | 70.42 ± 0.41 | 63.85 ± 1.40  | 46.59 ± 22.80 | 60.29          |
> | **Ours**               | **93.04 ± 0.24** | **90.69 ± 0.24** | **88.90 ± 0.15** | **90.88**  | **71.06 ± 0.22** | **64.61 ± 0.39** | **57.17 ± 0.80** | **64.28**  |

---

> > ### Author Response · Authors · 2024-08-16
> >
> > > Reproducibility and experimental details
> >
> > - For training on Imagenet, we use the setup similar to [ViT paper](https://openaccess.thecvf.com/content/ICCV2021/papers/Yuan_Tokens-to-Token_ViT_Training_Vision_Transformers_From_Scratch_on_ImageNet_ICCV_2021_paper.pdf), and the following codebase: [link](https://github.com/google-research/big_vision/tree/main). This uses the Adam optimizer with a learning rate of 1e-3, and a weight decay of 0.0001. We found RGD to work with the same learning rate and weight decay setup with just tuning of the $\tau$ parameter.
> >
> > For experiments on meta-learning, we follow a similar setup as the following codebase: [link](https://github.com/RamnathKumar181/Task-Diversity-meta-learning). For RGD, we tune both learning rate and $\tau$ parameters with the same grid-search space as mentioned in Appendix D.1.
> >
> > For experiments on CIFAR, we follow the same setup as [meta-weight-net](https://github.com/xjtushujun/meta-weight-net), and tune the same parameters as described in Appendix D.1.
> >
> > For MET, we used the same setup as the [MET paper](https://arxiv.org/abs/2206.08564), and tune the learning rate, and $tau$ parameter with the same hyperparameter space as described in Appendix D.1. We obtained this code through contacting one of the authors, and we believe the code might not be publicly available yet.
> >
> > To ensure fair comparisons, we integrated RGD into existing baseline codebase for each of the experiments, maintaining the same optimizer (primarily Adam for most experiments, SGD for CIFAR experiments) for both baseline and RGD versions.  RGD introduces only one additional hyperparameter, the clipping factor ($\tau$).  The optimizer, weight decay, batch sizes, etc. were kept constant with the baseline across all our experiments. We hope this helps other practitioners replicate our code, and help alleviate any issues around reproducibility.

---

> > ### Author Response · Authors · 2024-08-16
> >
> > > Comparison with TERM
> > - We agree that TERM and ABSGD are closely related methods, which is why we have had a detailed discussion with these techniques in our paper (see table 1 and also appendix A).  The key differences between TERM/ABSGD and RGD are: (1) weight clipping in RGD for outlier robustness and (2) the direct optimization of the inner objective in RGD, which avoids the need for maintaining moving averages in TERM/ABSGD (this reduces one hyper-parameter in our algorithm, and makes it easy to use in practice).  We will make necessary changes to the paper to reflect this comparison.
> >
> > > Other related work
> > - We will add a discussion of Superquantile approaches to the related work section, citing the suggested references. We acknowledge that these methods have not yet been scaled to the same level as our work, but they represent an interesting direction for future research.

---

> ### Comment · Reviewer_KKmY · 2024-08-19
>
> I would like to thank the authors for their detailed response. Their clarifications regarding SAM, section 3.1, and robustness to outliers fully address my questions. The provided experimental details, if added to the manuscript, will alleviate my concerns about reproducibility.
> A minor detail, though: please ensure that all published works are referenced correctly. It seems to me that there are several instances where published works have been erroneously referenced as preprints, for example:
>
> Sinha, Aman, Hongseok Namkoong, and John Duchi. "Certifying Some Distributional Robustness with Principled Adversarial Training." -> ICLR 2018
>
> Zhu, Deyao, et al. "Minigpt-4: Enhancing vision-language understanding with advanced large language models." -> ICLR 2024
>
> Ramnath Kumar, Tristan Deleu, and Yoshua Bengio. The effect of diversity in meta-learning.  -> AAAI 2023
>
> Ishaan Gulrajani and David Lopez-Paz. In search of lost domain generalization -> ICLR 2021
>
> Diederik P Kingma and Jimmy Ba. Adam: A method for stochastic optimization. -> ICLR 2015
>
> Anastasia Koloskova, Hadrien Hendrikx, and Sebastian U Stich. Revisiting gradient clipping: Stochastic bias
> and tight convergence guarantees. -> ICLR 2023
>
> Zhaoqi Leng, Mingxing Tan, Chenxi Liu, Ekin Dogus Cubuk, Xiaojie Shi, Shuyang Cheng, and Dragomir
> Anguelov. Polyloss: A polynomial expansion perspective of classification loss functions. -> ICLR 2022
>
> and please check the rest.

---

> > ### Author Response · Authors · 2024-08-20
> >
> > We thank the reviewer for their detailed feedback. We will incorporate it in our final manuscript.

---

### Decision · Action_Editor_9md6 · 2024-09-10

**Recommendation:** Accept with minor revision

**Comment:**

The reviewers appreciated the exposition of the method; reviewer `KKmY` had doubt about comparison with other methods and reproducibility, that were cleared by the authors' response.
All those should be incorporated in the revision, together with the confidence intervals required by `9md6`.

**Audience:**

Yes

**Claims And Evidence:**

The reviewers have appreciated the support for the claims and evidence made in the paper, in particular `sqSP` appreciated the experimental validation.